# PDE-DRIVEN SPATIOTEMPORAL DISENTANGLEMENT

**Jérémie Donà,**[†][*] **Jean-Yves Franceschi,**[†][*] **Sylvain Lamprier**[†] **& Patrick Gallinari**[†][‡]
[†]Sorbonne Université, CNRS, LIP6, F-75005 Paris, France
[‡]Criteo AI Lab, Paris, France
`firstname.lastname@lip6.fr`

## ABSTRACT

A recent line of work in the machine learning community addresses the problem of predicting high-dimensional spatiotemporal phenomena by leveraging specific tools from the differential equations theory. Following this direction, we propose in this article a novel and general paradigm for this task based on a resolution method for partial differential equations: the separation of variables. This inspiration allows us to introduce a dynamical interpretation of spatiotemporal disentanglement. It induces a principled model based on learning disentangled spatial and temporal representations of a phenomenon to accurately predict future observations. We experimentally demonstrate the performance and broad applicability of our method against prior state-of-the-art models on physical and synthetic video datasets.

## 1 INTRODUCTION

The interest of the machine learning community in physical phenomena has substantially grown for the last few years (Shi et al., 2015; Long et al., 2018; Greydanus et al., 2019). In particular, an increasing amount of works studies the challenging problem of modeling the evolution of dynamical systems, with applications in sensible domains like climate or health science, making the understanding of physical phenomena a key challenge in machine learning. To this end, the community has successfully leveraged the formalism of dynamical systems and their associated differential formulation as powerful tools to specifically design efficient prediction models. In this work, we aim at studying this prediction problem with a principled and general approach, through the prism of Partial Differential Equations (PDEs), with a focus on learning spatiotemporal disentangled representations.

Prediction via spatiotemporal disentanglement was first studied in video prediction works, in order to separate static and dynamic information (Denton & Birodkar, 2017) for prediction and interpretability purposes. Existing models are particularly complex, involving either adversarial losses or variational inference. Furthermore, their reliance on Recurrent Neural Networks (RNNs) hinders their ability to model spatiotemporal phenomena (Yıldız et al., 2019; Ayed et al., 2020; Franceschi et al., 2020). Our proposition addresses these shortcomings with a simplified and improved model by grounding spatiotemporal disentanglement in the PDE formalism.

Spatiotemporal phenomena obey physical laws such as the conservation of energy, that lead to describe the evolution of the system through PDEs. Practical examples include the conservation of energy for physical systems (Hamilton, 1835), or the equation describing constant illumination in a scene (Horn & Schunck, 1981) for videos that has had a longstanding impact in computer vision with optical flow methods (Dosovitskiy et al., 2015; Finn et al., 2016). We propose to model the evolution of partially observed spatiotemporal phenomena with unknown dynamics by leveraging a formal method for the analytical resolution of PDEs: the functional separation of variables (Miller, 1988). Our framework formulates spatiotemporal disentanglement for prediction as learning a separable solution, where spatial and dynamic information are represented in separate variables. Besides offering a novel interpretation of spatiotemporal disentanglement, it confers simplicity and performance compared to existing methods: disentanglement is achieved through the sole combination of a prediction objective and regularization penalties, and the temporal dynamics is defined by a learned Ordinary Differential Equation (ODE). We experimentally demonstrate the applicability, disentanglement capacity and

---

[*]Equal contribution.

forecasting performance of the proposed model on various spatiotemporal phenomena involving standard physical processes and synthetic video datasets against prior state-of-the-art models.

## 2    RELATED WORK

Our contribution deals with two main directions of research: spatiotemporal disentanglement and the coupling of neural networks and PDEs.

**Spatiotemporal disentanglement.**    Disentangling factors of variations is an essential representation learning problem (Bengio et al., 2013). Its cardinal formulation for static data has been extensively studied, with state-of-the-art solutions (Locatello et al., 2019) being essentially based on Variational Autoencoders (VAEs; Kingma & Welling, 2014; Rezende et al., 2014). As for sequential data, several disentanglement notions have been formulated, ranging from distinguishing objects in a video (Hsieh et al., 2018; van Steenkiste et al., 2018) to separating and modeling multi-scale dynamics (Hsu et al., 2017; Yingzhen & Mandt, 2018).

We focus in this work on the dissociation of the dynamics and visual aspects for spatiotemporal data. Even in this case, dissociation can take multiple forms. Examples in the video generation community include decoupling the foreground and background (Vondrick et al., 2016), constructing structured frame representations (Villegas et al., 2017b; Minderer et al., 2019; Liu et al., 2019), extracting physical dynamics (Le Guen & Thome, 2020), or latent modeling of dynamics in a state-space manner (Fraccaro et al., 2017; Franceschi et al., 2020). Closer to our work, Denton & Birodkar (2017), Villegas et al. (2017a) and Hsieh et al. (2018) introduced in their video prediction models explicit latent disentanglement of static and dynamic information obtained using adversarial losses (Goodfellow et al., 2014) or VAEs. Disentanglement has also been introduced in more restrictive models relying on data-specific assumptions (Kosiorek et al., 2018; Jaques et al., 2020), and in video generation (Tulyakov et al., 2018). We aim in this work at grounding and improving spatiotemporal disentanglement with more adapted inductive biases by introducing a paradigm leveraging the functional separation of variables resolution method for PDEs.

**Spatiotemporal prediction and PDE-based neural network models.**    An increasing number of works combining neural networks and differential equations for spatiotemporal forecasting have been produced for the last few years. Some of them show substantial improvements for the prediction of dynamical systems or videos compared to standard RNNs by defining the dynamics using learned ODEs (Rubanova et al., 2019; Yıldız et al., 2019; Ayed et al., 2020; Le Guen & Thome, 2020), following Chen et al. (2018), or adapting them to stochastic data (Ryder et al., 2018; Li et al., 2020; Franceschi et al., 2020). Most PDE-based spatiotemporal models exploit some prior physical knowledge. It can induce the structure of the prediction function (Brunton et al., 2016; de Avila Belbute-Peres et al., 2018) or specific cost functions, thereby improving model performances. For instance, de Bézenac et al. (2018) shape their prediction function with an advection-diffusion mechanism, and Long et al. (2018; 2019) estimate PDEs and their solutions by learning convolutional filters proven to approximate differential operators. Greydanus et al. (2019), Chen et al. (2020) and Toth et al. (2020) introduce non-regression losses by taking advantage of Hamiltonian mechanics (Hamilton, 1835), while Tompson et al. (2017) and Raissi et al. (2020) combine physically inspired constraints and structural priors for fluid dynamic prediction. Our work deepens this literature by establishing a novel link between a resolution method for PDEs and spatiotemporal disentanglement, thereby introducing a data-agnostic model leveraging any static information in observed phenomena.

## 3    BACKGROUND: SEPARATION OF VARIABLES

Solving high-dimensional PDEs is a difficult analytical and numerical problem (Bungartz & Griebel, 2004). Variable separation aims at simplifying it by decomposing the solution, e.g., as a simple combination of lower-dimensional functions, thus reducing the PDE to simpler differential equations.

### 3.1    SIMPLE CASE STUDY

Let us introduce this technique through a standard application, with proofs in Appendix A.1, on the one-dimensional heat diffusion problem (Fourier, 1822), consisting in a bar of length $L$, whose

temperature at time $t$ and position $x$ is denoted by $u(x, t)$ and satisfies:

$$\frac{\partial u}{\partial t} = c^2 \frac{\partial^2 u}{\partial x^2}, \qquad u(0, t) = u(L, t) = 0, \qquad u(x, 0) = f(x). \qquad (1)$$

Suppose that a solution $u$ is product-separable, i.e., it can be decomposed as: $u(x, t) = u_1(x) \cdot u_2(t)$. Combined with Equation (1), it leads to $c^2 u_1''(x)/u_1(x) = u_2'(t)/u_2(t)$. The left- and right-hand sides of this equation are respectively independent from $t$ and $x$. Therefore, both sides are constant, and solving both resulting ODEs gives solutions of the form, with $\mu \in \mathbb{R}$ and $n \in \mathbb{N}$:

$$u(x, t) = \mu \sin(n\pi x/L) \times \exp\left(-\left(cn\pi/L\right)^2 t\right). \qquad (2)$$

The superposition principle and the uniqueness of solutions under smoothness constraints allow then to build the set of solutions of Equation (1) with linear combinations of separable solutions (Le Dret & Lucquin, 2016). Besides this simple example, separation of variables can be more elaborate.

## 3.2 FUNCTIONAL SEPARATION OF VARIABLES

The functional separation of variables (Miller, 1988) generalizes this method. Let $u$ be a function obeying a given arbitrary PDE. The functional variable separation method amounts to finding a parameterization $z$, a functional $U$, an entangling function $\xi$, and representations $\phi$ and $\psi$ such that:

$$z = \xi\big(\phi(x), \psi(t)\big), \qquad u(x, t) = U(z). \qquad (3)$$

Trivial choices $\xi = u$ and identity function as $U$, $\phi$ and $\psi$ ensure the validity of this reformulation. Finding suitable $\phi$, $\psi$, $U$, and $\xi$ with regards to the initial PDE can facilitate its resolution by inducing separate simpler PDEs on $\phi$, $\psi$, and $U$. For instance, product-separability is retrieved with $U = \exp$. General results on the existence of separable solutions have been proven (Miller, 1983), though their uniqueness depends on the initial conditions and the choice of functional separation (Polyanin, 2020).

Functional separation of variables finds broad applications. It helps to solve refinements of the heat equation, such as generalizations with an advection term (see Appendix A.2) or with complex diffusion and source terms forming a general transport equation (Jia et al., 2008). Besides the heat equation, functional separation of PDEs is also applicable in various physics fields like reaction-diffusion with non-linear sources or convection-diffusion phenomena (Polyanin, 2019; Polyanin & Zhurov, 2020), Hamiltonian physics (Benenti, 1997), or even general relativity (Kalnins et al., 1992).

Reparameterizations such as Equation (3) implement a separation of spatial and temporal factors of variations, i.e., spatiotemporal disentanglement. We introduce in the following a learning framework based on this general method.

## 4 PROPOSED METHOD

We propose to model spatiotemporal phenomena using the functional variable separation formalism. We first describe our notations and then derive a principled model and constraints from this method.

### 4.1 PROBLEM FORMULATION THROUGH SEPARATION OF VARIABLES

We consider a distribution $\mathcal{P}$ of observed spatiotemporal trajectories and corresponding observation samples $v = (v_{t_0}, v_{t_0 + \Delta t}, \ldots, v_{t_1})$, with $v_t \in \mathcal{V} \subseteq \mathbb{R}^m$ and $t_1 = t_0 + \nu \Delta t$. Each sequence $v \sim \mathcal{P}$ corresponds to an observation of a dynamical phenomenon, assumed to be described by a hidden functional $u_v$ (also denoted by $u$ for the sake of simplicity) of space coordinates $x \in \mathcal{X} \subseteq \mathbb{R}^s$ and time $t \in \mathbb{R}$ that characterizes the trajectories. More precisely, $u_v$ describes an unobserved continuous dynamics and $v$ corresponds to instantaneous discrete spatial measurements associated to this dynamics. Therefore, we consider that $v_t$ results from a time-independent function $\zeta$ of the mapping $u_v(\cdot, t)$. For example, $v$ might consist in temperatures measured at some points of the sea surface, while $u_v$ would be the complete ocean circulation model. In other words, $v$ provides a partial information about $u_v$ and is a projection of the full dynamics. We seek to learn a model which, when conditioned on prior observations, can predict future observations.

To this end, we posit that the state $u$ of each observed trajectory $v$ is driven by a hidden PDE, shared among all trajectories; we discuss this assumption in details in Appendix C.1. Learning such a PDE

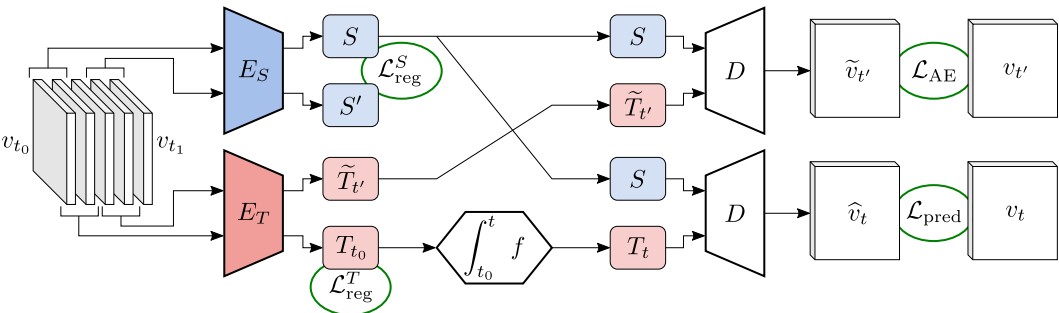

Figure 1: Computational graph of the proposed model. $E_S$ and $E_T$ take contiguous observations as input; time invariance is enforced on $S$; the evolution of $T_t$ is modeled with an ODE and is constrained to coincide with $E_T$; $T_{t_0}$ is regularized; forecasting amounts to decoding from $S$ and $T_t$.

and its solutions would then allow us to model observed trajectories $v$. However, directly learning solutions to high-dimensional unknown PDEs is a complex task (Bungartz & Griebel, 2004; Sirignano & Spiliopoulos, 2018). We aim in this work at simplifying this resolution. We propose to do so by relying on the functional separation of variables of Equation (3), in order to leverage a potential separability of the hidden PDE. Therefore, analogously to Equation (3), we propose to formulate the problem as learning observation-constrained $\phi$, $\psi$ and $U$, as well as $\xi$ and $\zeta$, such that:

$$z = \xi\big(\phi(x), \psi(t)\big), \qquad u(x,t) = U(z), \qquad v_t = \zeta\big(u(\cdot, t)\big), \qquad (4)$$

with $\phi$ and $\psi$ allowing to disentangle the prediction problem. In the formalism of the functional separation of variables, this amounts to decomposing the full solution $u$, thereby learning a spatial PDE on $\phi$, a temporal ODE on $\psi$, and a PDE on $U$, as well as their respective solutions.

## 4.2 FUNDAMENTAL LIMITS AND RELAXATION

Directly learning $u$ is, however, a restrictive choice. Indeed, when formulating PDEs such as in Equation (1), spatial coordinates ($x$, $y$, etc.) and time $t$ appear as variables of the solution. Yet, unlike in fully observable phenomena studied by Sirignano & Spiliopoulos (2018) and Raissi (2018), directly accessing theses variables in practice can be costly or infeasible in our partially observed setting. In other words, the nature and number of these variables are unknown. For example, the dynamic of the observed sea surface temperature is highly dependent on numerous unobserved variables such as temperature at deeper levels or wind intensity. Explicitly taking into account these unobserved variables can only be done with prior domain knowledge. To maintain the generality of the proposed approach, we choose not to make any data-specific assumption on these unknown variables.

We overcome these issues by eliminating the explicit modeling of spatial coordinates by learning dynamic and time-invariant representations accounting respectively for the time-dependent and space-dependent parts of the solution. Indeed, Equation (4) induces that these spatial coordinates, hence the explicit resolution of PDEs on $u$ or $U$, can be ignored, as it amounts to learning $\phi$, $\psi$ and $D$ such that:

$$v_t = (\zeta \circ U \circ \xi)\big(\phi(\cdot), \psi(t)\big) = D\big(\phi, \psi(t)\big). \qquad (5)$$

In order to manipulate functionals $\phi$ and $\psi$ in practice, we respectively introduce learnable time-invariant and time-dependent representations of $\phi$ and $\psi$, denoted by $S$ and $T$, such that:

$$\phi \equiv S \in \mathcal{S} \subseteq \mathbb{R}^d, \qquad \psi \equiv T : t \mapsto T_t \in \mathcal{T} \subseteq \mathbb{R}^p, \qquad (6)$$

where the dependence of $\psi \equiv T$ on time $t$ will be modeled using a temporal ODE following the separation of variables, and the function $\phi$, and consequently its spatial PDE, are encoded into a vectorial representation $S$. Besides their separation of variables basis, the purpose of $S$ and $T$ is to capture spatial and motion information of the data. For instance, $S$ could encode static information such as objects appearance, while $T$ typically contains motion variables.

$S$ and $T_{t_0}$, because of their dependence on $v$ in Equations (5) and (6), are inferred from an observation history, or conditioning frames, $V_\tau(t_0)$, where $V_\tau(t) = (v_t, v_{t+\Delta t}, \ldots, v_{t+\tau\Delta t})$, using respectively encoder networks $E_S$ and $E_T$. We parameterize $D$ of Equation (5) as a neural network that acts on both $S$ and $T_t$, and outputs the estimated observation $\widehat{v}_t = D(S, T_t)$. Unless specified otherwise, $S$ and $T_t$ are fed concatenated into $D$, which then learns the parameterization $\xi$ of their combination.

### 4.3 TEMPORAL ODE

The separation of variables allows us to partly reduce the complex task of learning and integrating PDEs to learning and integrating an ODE on $\psi$, which has been extensively studied in the literature, as explained in Section 2. We therefore model the evolution of $T_t$, thereby the dynamics of our system, with a first-order ODE:

$$\frac{\partial T_t}{\partial t} = f(T_t) \qquad\qquad \Leftrightarrow \qquad\qquad T_t = T_{t_0} + \int_{t_0}^{t} f(T_{t'})\, \mathrm{d}t' \qquad (7)$$

Note that the first-order ODE assumption can be taken without loss of generality since any ODE is equivalent to a higher-dimensional first-order ODE. Following Chen et al. (2018), $f$ is implemented by a neural network and Equation (7) is solved with an ODE resolution scheme. Suppose initial ODE conditions $S$ and $T_{t_0}$ have been computed with $E_S$ and $E_T$. This leads to the following simple forecasting scheme, enforced by the corresponding regression loss:

$$\widehat{v}_t = D\left(S, T_{t_0} + \int_{t_0}^{t} f(T_{t'})\, \mathrm{d}t'\right), \qquad \mathcal{L}_{\text{pred}} = \frac{1}{\nu+1}\sum_{i=0}^{\nu}\frac{1}{m}\|\widehat{v}_{t_0+i\Delta t} - v_{t_0+i\Delta t}\|_2^2, \qquad (8)$$

where $\nu + 1$ is the number of observations, and $m$ is the dimension of the observed variables $v$.

Equation (8) ensures that the evolution of $T$ is coherent with the observations; we should enforce its consistency with $E_T$. Indeed, the dynamics of $T_t$ is modeled by Equation (7), while only its initial condition $T_{t_0}$ is computed with $E_T$. However, there is no guaranty that $T_t$, computed via integration, matches $E_T(V_\tau(t))$ at any other time $t$, while they should in principle coincide. We introduce the following autoencoding constraint mitigating their divergence, thereby stabilizing the evolution of $T$:

$$\mathcal{L}_{\text{AE}} = \frac{1}{m}\left\|D\Big(S, E_T\big(V_\tau(t_0+i\Delta t)\big)\Big) - v_{t_0+i\Delta t}\right\|_2^2, \qquad \text{with } i \sim \mathcal{U}\big([\![0, \nu-\tau]\!]\big). \qquad (9)$$

### 4.4 SPATIOTEMPORAL DISENTANGLEMENT

As indicated hereinabove, the spatial PDE on $\phi$ is assumed to be encoded into $S$. Nonetheless, since $S$ is inferred from an observation history, we need to explicitly enforce its time independence. In the PDE formalism, this is equivalent to:

$$\frac{\partial E_S\big(V_\tau(t)\big)}{\partial t} = 0 \qquad\qquad \Leftrightarrow \qquad\qquad \int_{t_0}^{t_1-\tau\Delta t}\left\|\frac{\partial E_S\big(V_\tau(t)\big)}{\partial t}\right\|_2^2 \mathrm{d}t = 0. \qquad (10)$$

However, enforcing Equation (10) raises two crucial issues. Firstly, in our partially observed setting, there can be variations of observable content, for instance when an object conceals another one. Therefore, strictly enforcing a null time derivative is not desirable as it prevents $E_S$ to extract accessible information that could be obfuscated in the sequence. Secondly, estimating this derivative in practice in our setting is unfeasible and costly because of the coarse temporal discretization of the data and the computational cost of $E_S$; see Appendix B for more details. We instead introduce a discretized penalty in our minimization objective, discouraging variations of content between two distant time steps, with $d$ being the dimension of $S$:

$$\mathcal{L}_{\text{reg}}^S = \frac{1}{d}\left\|E_S\big(V_\tau(t_0)\big) - E_S\big(V_\tau(t_1-\tau\Delta t)\big)\right\|_2^2. \qquad (11)$$

It allows us to overcome the previously stated issues by not enforcing a strict invariance of $S$ and removing the need to estimate any time derivative. Note that this formulation actually originates from Equation (10) using the Cauchy-Schwarz inequality (see Appendix B for a more general derivation).

Abstracting the spatial ODE on $\phi$ from Equation (4) into a generic representation $S$ leads, without additional constraints, to an underconstrained problem where spatiotemporal disentanglement cannot be guaranteed. Indeed, $E_S$ can be set to zero to satisfy Equation (11) without breaking any prior constraint, because static information is not prevented to be encoded into $T$. Accordingly, information in $S$ and $T$ needs to be segmented.

Thanks to the design of our model, it suffices to ensure that $S$ and $T$ are disentangled at initial time $t_0$ for them be to be disentangled at all $t$. Indeed, the mutual information between two variables

is preserved by invertible transformations. Equation (7) is an ODE and $f$, as a neural network, is Lipschitz-continuous, so the ODE flow $T_t \mapsto T_{t'}$ is invertible. Therefore, disentanglement between $S$ and $T_t$, characterized by a low mutual information between both variables, is preserved through time; see Appendix C for a detailed discussion. We thus only constrain the information quantity in $T_{t_0}$ by using a Gaussian prior to encourage it to exclusively contain necessary dynamic information:

$$\mathcal{L}_{\text{reg}}^T = \frac{1}{p}\|T_{t_0}\|_2^2 = \frac{1}{p}\left\|E_T\left(V_\tau(t_0)\right)\right\|_2^2. \tag{12}$$

### 4.5 Loss Function

The minimized loss is a linear combination of Equations (8), (9), (11) and (12):

$$\mathcal{L}(v) = \mathbb{E}_{v \sim \mathcal{P}}\left[\lambda_{\text{pred}}\mathcal{L}_{\text{pred}} + \lambda_{\text{AE}} \cdot \mathcal{L}_{\text{AE}} + \lambda_{\text{reg}}^S \cdot \mathcal{L}_{\text{reg}}^S + \lambda_{\text{reg}}^T \cdot \mathcal{L}_{\text{reg}}^T\right], \tag{13}$$

as illustrated in Figure 1. In the following, we conventionally set $\Delta t = 1$. Note that the presented approach could be generalized to irregularly sampled observation times thanks to the dedicated literature (Rubanova et al., 2019), but this is out of the scope of this paper.

## 5 Experiments

We study in this section the experimental results of our model on various spatiotemporal phenomena with physical, synthetic video and real-world datasets, which are briefly presented in this section and in more details in Appendix D. We demonstrate the relevance of our model with ablation studies and its performance by comparing it with more complex state-of-the-art models. Performances are assessed thanks to standard metrics (Denton & Fergus, 2018; Le Guen & Thome, 2020) Mean Squared Error (MSE, lower is better) or its alternative Peak Signal-to-Noise Ratio (PSNR, higher is better), and Structured Similarity (SSIM, higher is better). We refer to Appendix F for more experiments and prediction examples, to Appendix E for training information and to the supplementary material for the corresponding code[1] and datasets.

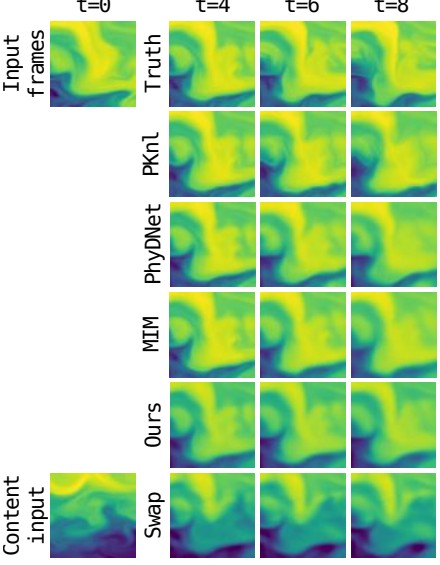

Figure 2: Example of predictions of compared models on SST. Content swap preserves the location of extreme temperature regions which determine the movement while modifying the magnitude of all regions, especially in temperate areas.

### 5.1 Physical Datasets: Wave Equation and Sea Surface Temperature

We first investigate two synthetic dynamical systems and a real-world dataset in order to show the advantage of PDE-driven spatiotemporal disentanglement for forecasting physical phenomena. To analyze our model, we first lean on the wave equation, occurring for example in acoustic or electromagnetism, with source term like Saha et al. (2020), to produce the WaveEq dataset consisting in $64 \times 64$ normalized images of the phenomenon. We additionally build the WaveEq-100 dataset by extracting 100 pixels, chosen uniformly at random and shared among sequences, from WaveEq frames; this experimental setting can be thought of as measurements from sensors partially observing the phenomenon. We also test and compare our model on the real-world dataset SST, derived from the data assimilation engine NEMO (Madec & Team) and introduced by de Bézenac et al. (2018), consisting in $64 \times 64$ frames showing the evolution of the sea surface temperature. Modeling its evolution is particularly challenging as its dynamic is highly non-linear, chaotic, and involves several unobserved quantities (e.g., forcing terms).

---

[1]Our source code is also publicly released at the following URL: https://github.com/JeremDona/spatiotemporal_variable_separation.

Table 1: Forecasting performance on WaveEq-100, WaveEq and SST of compared models with respect to indicated prediction horizons. Bold scores indicate the best performing method.

| Models | WaveEq-100 | WaveEq | SST | | | |
|---|---|---|---|---|---|---|
| | MSE | | MSE | | SSIM | |
| | $t+40$ | $t+40$ | $t+6$ | $t+10$ | $t+6$ | $t+10$ |
| PKnl | — | — | 1.28 | 2.03 | 0.6686 | 0.5844 |
| PhyDNet | — | — | 1.27 | 1.91 | 0.5782 | 0.4645 |
| SVG | — | — | 1.51 | 2.06 | 0.6259 | 0.5595 |
| MIM | — | — | 0.91 | 1.45 | 0.7406 | 0.6525 |
| Ours | $\mathbf{4.33 \times 10^{-5}}$ | $\mathbf{1.44 \times 10^{-4}}$ | **0.86** | **1.43** | **0.7466** | **0.6577** |
| Ours (without $S$) | $1.33 \times 10^{-4}$ | $5.09 \times 10^{-4}$ | 0.95 | 1.50 | 0.7204 | 0.6446 |

Table 2: Prediction and content swap scores of all compared models on Moving MNIST. Bold scores indicate the best performing method.

| Models | Pred. $(t+10)$ | | Pred. $(t+95)$ | | Swap $(t+10)$ | | Swap $(t+95)$ | |
|---|---|---|---|---|---|---|---|---|
| | PSNR | SSIM | PSNR | SSIM | PSNR | SSIM | PSNR | SSIM |
| SVG | 18.18 | 0.8329 | 12.85 | 0.6185 | — | — | — | — |
| MIM | **24.16** | 0.9113 | 16.50 | 0.6529 | — | — | — | — |
| DrNet | 14.94 | 0.6596 | 12.91 | 0.5379 | 14.12 | 0.6206 | 12.80 | 0.5306 |
| DDPAE | 21.17 | 0.8814 | 13.56 | 0.6446 | **18.44** | 0.8256 | 13.25 | 0.6378 |
| PhyDNet | 23.12 | **0.9128** | 16.46 | 0.3878 | 12.04 | 0.5572 | 13.49 | 0.2839 |
| Ours | 21.70 | 0.9088 | **17.50** | **0.7990** | 18.42 | **0.8368** | **16.50** | **0.7713** |

We compare our model on these three datasets to its alternative version with $S$ removed and integrated into $T$, thus also removing $\mathcal{L}_{\text{reg}}^{S}$ and $\mathcal{L}_{\text{reg}}^{T}$. We also include the state-of-the-art PhyDNet (Le Guen & Thome, 2020), MIM (Wang et al., 2019b), SVG (Denton & Fergus, 2018) and SST-specific PKnl (de Bézenac et al., 2018) in the comparison on SST; only PhyDNet and PKnl were originally tested on this dataset by their authors. Results are compiled in Table 1 and an example of prediction is depicted in Figure 2.

On these three datasets, our model produces more accurate long-term predictions with $S$ than without it. This indicates that learning an invariant component facilitates training and improves generalization. The influence of $S$ can be observed by replacing the $S$ of a sequence by another one extracted from another sequence, changing the aspect of the prediction, as shown in Figure 2 (swap row). We provide in Appendix F further samples showing the influence of $S$ in the prediction. Even though there is no evidence of intrinsic separability in SST, our trained algorithm takes advantage of its time-invariant component. Indeed, our model outperforms PKnl despite the data-specific structure of the latter, the stochastic SVG and the high-capacities PhyDNet and MIM model, whereas removing its static component suppresses our advantage.

We highlight that MIM is a computationally-heavy model that manipulates in an autoregressive way 64 times larger latent states than ours, hence its better reconstruction ability at the first time step. However, its sharpness and movement gradually vanish, explaining its lower performance than ours. We refer to Appendix F.3 for additional discussion on the application of our method and its performance on SST.

## 5.2 A SYNTHETIC VIDEO DATASET: MOVING MNIST

We also assess the prediction and disentanglement performance of our model on the Moving MNIST dataset (Srivastava et al., 2015) involving MNIST digits (LeCun et al., 1998) bouncing over frame borders. This dataset is particularly challenging in the literature for long-term prediction tasks. We

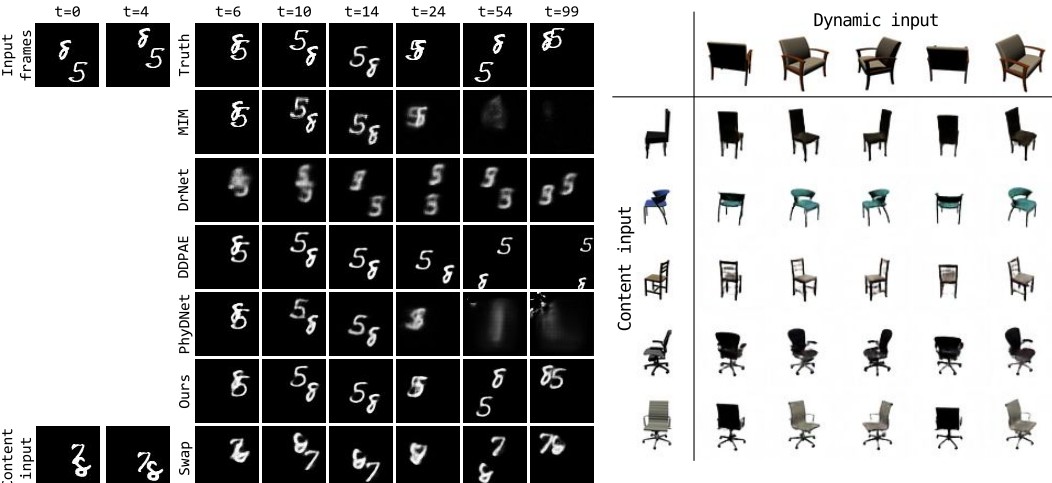

Figure 3: Predictions of compared models on Moving MNIST, and content swap experiment for our model.

Figure 4: Fusion of content (first column) and dynamic (first row) variables in our model on 3D Warehouse Chairs.

compare our model to competitive baselines: the non-disentangled SVG (Denton & Fergus, 2018) and MIM (Wang et al., 2019b), as well as forecasting models with spatiotemporal disentanglement ablities DrNet (Denton & Birodkar, 2017), DDPAE (Hsieh et al., 2018) and PhyDNet. We highlight that all these models leverage powerful machine learning tools such as adversarial losses, VAEs and high-capacity temporal architectures, whereas ours is solely trained using regression penalties and small-size latent representations. We perform as well a full ablation study of our model to confirm the relevance of the introduced method.

Results reported in Table 2 and illustrated in Figure 3 correspond to two tasks: prediction and disentanglement, at both short and long-term horizons. Disentanglement is evaluated via content swapping, which consists in replacing the content representation of a sequence by the one of another sequence, which should result for a perfectly disentangled model in swapping digits of both sequences. This is done by taking advantage of the synthetic nature of this dataset that allows us to implement the ground truth content swap and compare it to the generated swaps of the model.

Reported results show the advantage of our model against all baselines. Long-term prediction challenges them as their performance and predictions collapse in the long run. This shows that the baselines, including high-capacity models MIM and PhyDNet that leverage powerful ConvLTSMs (Shi et al., 2015), have difficulties separating content and motion. Indeed, a model separating correctly content and motion should maintain digits appearance even when it miscalculates their trajectories, like DDPAE which alters only marginally the digits in Figure 3. In contrast, ours manages to produce consistent samples even at $t + 95$, making it reach state-of-the-art performance. Moreover, we significantly outperform all baselines in the content swap experiment, showing the clear advantage of the proposed PDE-inspired simple model for spatiotemporally disentangled prediction.

Ablation studies developed in Table 4 confirm that this advantage is due to the constraints motivated by the separation of variables. Indeed, the model without $S$ fails at long-term forecasting, and removing any non-prediction penalty of the training loss substantially harms performances. In particular, the invariance loss on the static component and the regularization of initial condition $T_{t_0}$ are essential, as their absence hinders both prediction and disentanglement. The auto-encoding constraint makes predictions more stable, allowing accurate long-term forecasting and disentanglement. This ablation study also confirms the necessity to constrain the $\ell_2$ norm of the dynamic variable (see Equation (12)) for the model to disentangle. Comparisons of Table 2 actually show that enforcing this loss on the first time step only is sufficient to ensure state-of-the-art disentanglement, as advocated in Section 4.4.

Finally, we assess whether the temporal ODE of Equation (7) induced by the separation of variables is advantageous by replacing the dynamic model with a standard GRU RNN (Cho et al., 2014). Results reported in Table 4 show substantially better prediction and disentanglement performance for the original model grounded on the separation of variables, indicating the relevance of our approach.

Table 3: Prediction MSE ($\times 100 \times 32 \times 32 \times 2$) of compared models on TaxiBJ, with best MSE highlighted in bold.

| Ours | Ours (without $S$) | PhyDNet | MIM | E3D | C. LSTM | PredRNN | ConvLSTM |
|------|---------|---------|-----|-----|---------|---------|----------|
| **39.5** | 43.7 | 41.9 | 42.9 | 43.2 | 44.8 | 46.4 | 48.5 |

## 5.3 A MULTI-VIEW DATASET: 3D WAREHOUSE CHAIRS

We perform an additional disentanglement experiment on the 3D Warehouse Chairs dataset introduced by Aubry et al. (2014). This dataset contains 1393 three-dimensional models of chairs seen under various angles. Since all chairs are observed from the same set of angles, this constitutes a multi-view dataset enabling quantitative disentanglement experiments. We create sequences from this dataset for our model by assembling adjacent views of each chair to simulate its rotation from right to left. We then evaluate the disentanglement properties of our model with the same content swap experiments as for Moving MNIST. It is similar to one of Denton & Birodkar (2017)'s experiments who qualitatively tested their model on a similar, but smaller, multi-view chairs dataset. We achieve 18.70 PSNR and 0.7746 SSIM on this task, outperforming DrNet which only reaches 16.35 PSNR and 0.6992 SSIM. This is corroborated by qualitative experiments in Figures 4 and 11. We highlight that the encoder and decoder architectures of both competitors are identical, validating our PDE-grounded framework for spatiotemporal disentanglement of complex three-dimensional shapes.

## 5.4 A CROWD FLOW DATASET: TAXIBJ

We finally study the performance of our spatiotemporal model on the real-world TaxiBJ dataset (Zhang et al., 2017), consisting in taxi traffic flow in Beijing monitored on a $32 \times 32$ grid with an observation every thirty minutes. It is highly structured as the flows are dependent on the infrastructures of the city, and complex since methods have to account for non-local dependencies and model subtle changes in the evolution of the flows. It is a standard benchmark in the spatiotemporal prediction community (Wang et al., 2019b; Le Guen & Thome, 2020).

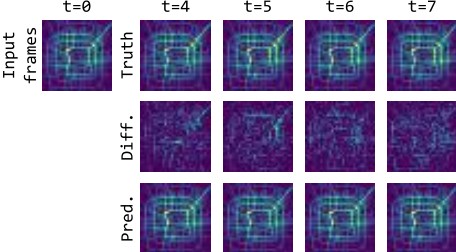

Figure 5: Example of ground truth and prediction of our model on TaxiBJ. The middle row shows the scaled difference between our predictions and the ground truth.

We compare our model in Table 3 against PhyDNet and MIM, as well as powerful baselines E3D-LSTM (E3D, Wang et al., 2019a), Causal LSTM (C. LSTM, Wang et al., 2018), PredRNN (Wang et al., 2017) and ConvLTSM (Shi et al., 2015), using results reported by Wang et al. (2019b) and Le Guen & Thome (2020). An example of prediction is given in Figure 5. We observe that we significantly overcome the state of the art on this complex spatiotemporal dataset. This improvement is notably driven by the disentanglement abilities of our model, as we observe in Table 3 that the alternative version of our model without $S$ achieves results comparable to E3D and worse than PhyDNet and MIM.

## 6 CONCLUSION

We introduce a novel method for spatiotemporal prediction inspired by the separation of variables PDE resolution technique that induces time invariance and regression penalties only. These constraints ensure the separation of spatial and temporal information. We experimentally demonstrate the benefits of the proposed model, which outperforms prior state-of-the-art methods on physical and synthetic video datasets. We believe that this work, by providing a dynamical interpretation of spatiotemporal disentanglement, could serve as the basis of more complex models further leveraging the PDE formalism. Another direction for future work could be extending the model with more involved tools such as VAEs to improve its performance, or adapt it to the prediction of natural stochastic videos (Denton & Fergus, 2018).

ACKNOWLEDGMENTS

We would like to thank all members of the MLIA team from the LIP6 laboratory of Sorbonne Université for helpful discussions and comments, as well as Vincent Le Guen for his help to reproduce PhyDNet results and process the TaxiBJ dataset.

We acknowledge financial support from the LOCUST ANR project (ANR-15-CE23-0027) and the European Union's Horizon 2020 research and innovation programme under grant agreement 825619 (AI4EU). This study has been conducted using E.U. Copernicus Marine Service Information. This work was granted access to the HPC resources of IDRIS under allocations 2020-AD011011360 and 2021-AD011011360R1 made by GENCI (Grand Equipement National de Calcul Intensif). Patrick Gallinari is additionally funded by the 2019 ANR AI Chairs program via the DL4CLIM project.

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

## A  PROOFS

### A.1  RESOLUTION OF THE HEAT EQUATION

In this section, we succinctly detail a proof for the existence and uniqueness for the solution to the two-dimensional heat equation. It shows that product-separable solutions allow to build the entire solution space for this problem, highlighting our interest in the research of separable solutions.

**Existence through separation of variables.**   Consider the heat equation problem:

$$\frac{\partial u}{\partial t} = c^2 \frac{\partial^2 u}{\partial x^2}, \qquad\qquad u(0,t) = u(L,t) = 0, \qquad\qquad u(x,0) = f(x). \qquad (14)$$

Assuming product separability of $u$ with $u(x,t) = u_1(x)u_2(t)$ in Equation (14) gives:

$$c^2 \frac{u_1''(x)}{u_1(x)} = \frac{u_2'(t)}{u_2(t)}. \qquad (15)$$

Both sides being independent of each other variables, they are equal to a constant denoted by $-\alpha$. If $\alpha$ is negative, solving the right side of Equation (15) results to non-physical solutions with exponentially increasing temperatures, and imposing border condition of Equation (14) makes this solution collapse to the null trivial solution. Therefore, we consider that $\alpha > 0$.

Both sides of Equation (15) being equal to a constant leads to a second-order ODE on $u_1$ and a first-order ODE on $u_2$, giving the solution shapes, with constants $A$, $B$ and $D$:

$$\begin{cases} u_1(x) & = A\cos\left(\sqrt{\alpha}x\right) + B\sin\left(\sqrt{\alpha}x\right) \\ u_2(t) & = D\mathrm{e}^{-\alpha c^2 t} \end{cases}. \qquad (16)$$

**Link with initial and boundary conditions.**   We now link the above equation to the boundary conditions of the problem. Because our separation is multiplicative, we can omit $D$ for non-trivial solutions and set it without loss of generality to $1$, as it only scales the values of $A$ and $B$.

Boundary condition $u(0,t) = u(L,t) = 0$, along with the fact that for all $t > 0$, $u_2(t) \neq 0$, give:

$$A = 0, \qquad\qquad\qquad B\mathrm{e}^{-\alpha c^2 t}\sin\left(\sqrt{\alpha}L\right) = 0, \qquad (17)$$

which means that, for a non-trivial solution (i.e., $B \neq 0$), we have for some $n \in \mathbb{N}$: $\sqrt{\alpha} = n\pi/L$. We can finally express our product-separable solution to the heat equation without initial conditions as:

$$u(x,t) = B\sin\left(\frac{n\pi}{L}x\right)\exp\left(-\left(\frac{cn\pi}{L}\right)^2 t\right). \qquad (18)$$

Considering the superposition principle, because the initial problem is homogeneous, all linear combinations of Equation (18) are solutions of the heat equation without initial conditions. Therefore, any following function is a solution of the heat equation without initial conditions.

$$u(x,t) = \sum_{n=0}^{+\infty} B_n \sin\left(\frac{n\pi}{L}x\right)\exp\left(-\left(\frac{cn\pi}{L}\right)^2 t\right). \qquad (19)$$

Finally, considering the initial condition $u(x,0) = f(x)$, a Fourier decomposition of $f$ allows to choose appropriate values for all coefficients $B_n$, showing that, for any initial condition $f$, there exists a solution to Equation (14) of the form of Equation (19).

**Uniqueness.**  We present here elements of proof for establishing the uniqueness of the solutions of Equation (14) that belong to $\mathcal{C}^2\big([0,1] \times \mathbb{R}_+\big)$. Detailed and rigorous proofs are given by Le Dret & Lucquin (2016).

The key element consists in establishing the so-called Maximum Principle which states that, considering a sufficiently smooth solution, the minimum value of the solution is reached on the boundary of the space and time domains.

For null border condition as in our case, this means that the norm of the solution $u$ is given by the norm of the initial condition $f$. Finally, let us consider two smooth solutions $U_1$ and $U_2$ of Equation (14). Then, their difference $v = U_1 - U_2$ follows the heat equation with null border and initial conditions (i.e, $v(x,0) = 0$). Because $v$ is as regular as $U_1$ and $U_2$, it satisfies the previous fact about the norm of the solutions, i.e, the norm of $v$ equals the norm of its initial condition: $\|v\| = 0$. Therefore, $v$ is null and so is $U_1 - U_2 = 0$, showing the uniqueness of the solutions.

Therefore, this shows that solutions of the form of Equation (19) shape the whole set of smooth solutions of Equation (14).

### A.2   Heat Equation with Advection Term

Consider the heat equation with a complementary advection term, for $x \in (-1,1)$, $t \in (0,T)$ and a constant $c \in \mathbb{R}_+$.

$$\frac{\partial u}{\partial t} + c\frac{\partial u}{\partial x} = \chi\frac{\partial^2 u}{\partial x^2}, \quad . \tag{20}$$

We give here details for the existence of product-separable solutions of Equation (20). To this end, let us choose real constants $\alpha$ and $\beta$, and consider the following change of variables for $u$:

$$u(x,t) = v(x,t)\mathrm{e}^{\alpha x + \beta t}. \tag{21}$$

The partial derivatives from Equation (20) can be rewritten as functions of the new variable $v$:

$$\frac{\partial u}{\partial t} = \frac{\partial v}{\partial t}\,\mathrm{e}^{\alpha x + \beta t} + \beta v\mathrm{e}^{\alpha x + \beta t} \tag{22}$$

$$\frac{\partial u}{\partial x} = \frac{\partial v}{\partial x}\,\mathrm{e}^{\alpha x + \beta t} + \alpha v\mathrm{e}^{\alpha x + \beta t} \tag{23}$$

$$\frac{\partial^2 u}{\partial x^2} = \frac{\partial^2 v}{\partial x^2}\,\mathrm{e}^{\alpha x + \beta t} + 2\alpha\,\frac{\partial v}{\partial x}\,\mathrm{e}^{\alpha x + \beta t} + \alpha^2 v\mathrm{e}^{\alpha x + \beta t} \tag{24}$$

Using these expressions in Equation (20) and dividing it by $\mathrm{e}^{\alpha x + \beta t}$ lead to:

$$\frac{\partial v}{\partial t} + \Big(\beta + c\alpha - \alpha^2\chi\Big)v + (c - 2\alpha\chi)\frac{\partial v}{\partial x} = \nu\frac{\partial^2 v}{\partial x^2}. \tag{25}$$

$\alpha$ and $\beta$ can then be set such that:

$$\beta + c\alpha - \alpha^2\chi = 0 \qquad\qquad c - 2\alpha\chi = 0, \tag{26}$$

to retrieve the standard two-dimensional heat equation of Equation (14) given by:

$$\frac{\partial v}{\partial t} = \chi\frac{\partial^2 v}{\partial x^2}, \tag{27}$$

which is known to have product-separable solutions as explained in the previous section. This more generally shows that all solutions of Equation (20) can be retrieved from solutions to Equation (14).

## B   Accessing Time Derivatives of $S$ and Deriving a Feasible Weaker Constraint

Explicitly constraining the time derivative of $E_S\big(V_\tau(t)\big)$ as explained in Section 4.4 is a difficult matter in practice. Indeed, $E_S$ does not take as input neither the time coordinate $t$ nor spatial coordinates $x$ and $y$ as done by Raissi (2018) and Sirignano & Spiliopoulos (2018), which allows

them to directly estimate the networks derivative thanks to automatic differentiation. In our case, $E_S$ rather takes as input a finite number of observations, making this derivative impractical to compute.

To discretize Equation (10) and find a weaker constraint, we chose to leverage the Cauchy-Schwarz inequality. We presented and used a version where we applied this inequality on the whole integration domain, i.e., from $t_0$ to $t_1 - \tau \Delta t$. We highlight that this inequality can also be applied on subintervals of the integration domain, generalizing our proposition. Indeed, let $p \in \mathbb{N}^*$ and consider a sequence of $t^{(k)}$ for $k \in [\![0, p]\!]$ such that $t_0 = t^{(0)} \leq t^{(1)} \leq \ldots \leq t^{(p)} = t_1 - \tau \Delta t$. Then, using the Cauchy-Schwarz inequality, we obtain:

$$
\begin{aligned}
\int_{t_0}^{t_1 - \tau \Delta t} \left\| \frac{\partial E_S(V_\tau(t))}{\partial t} \right\|_2^2 \mathrm{d}t &= \sum_{k=0}^{k=p} \int_{t^{(k-1)}}^{t^{(k)}} \left\| \frac{\partial E_S(V_\tau(t))}{\partial t} \right\|_2^2 \mathrm{d}t \\
&\geq \sum_{k=0}^{k=p} \frac{1}{t^{(k)} - t^{(k-1)}} \left\| \int_{t^{(k-1)}}^{t^{(k)}} \frac{\partial E_S(V_\tau(t))}{\partial t} \mathrm{d}t \right\|_2^2 \\
&\geq \sum_{k=0}^{k=p} \frac{1}{t^{(k)} - t^{(k-1)}} \left\| E_S\left(V_\tau\left(t^{(k)}\right)\right) - E_S\left(V_\tau\left(t^{(k-1)}\right)\right) \right\|_2^2 .
\end{aligned}
$$
(28)

Our constraint is a special case of this development, with $p = 1$. Nevertheless, we experimentally found that our simple penalty is sufficient to achieve state-of-the-art performance at a substantially reduced computational cost. We notice that other invariance constraints such as the one of Denton & Birodkar (2017) can also be derived thanks to framework, showing the generality of our approach.

## C   OF SPATIOTEMPORAL DISENTANGLEMENT

### C.1   MODELING SPATIOTEMPORAL PHENOMENA WITH DIFFERENTIAL EQUATIONS

Besides their increasing popularity to model spatiotemporal phenomena (see Section 2), the ability of residual networks to facilitate learning (Haber & Ruthotto, 2017) as well as the success of their continuous counterpart (Chen et al., 2018) motivate our choice. Indeed, learning ODEs or discrete approximations as residual networks has become standard for a variety of tasks such as classification, inpainting, and generative models. Consequently, their application to forecasting physical processes and videos is only a natural extension of its already broad applicability discussed in Section 2. Furthermore, they present interesting properties, as detailed below.

### C.2   SEPARATION OF VARIABLES PRESERVES THE MUTUAL INFORMATION OF S AND T THROUGH TIME

#### C.2.1   INVERTIBLE FLOW OF AN ODE

We first highlight that the general ODE Equation (7) admits, according to the Cauchy–Lipschitz theorem, exactly one solution for a given initial condition, since $f$ is implemented with a standard neural network (see Appendix E), making it Lipschitz-continuous. Consequently, the flow of this ODE, denoted by $\Phi_t$ and defined as:

$$
\Phi : \mathbb{R} \times \mathbb{R}^p \to \mathbb{R}^p
$$
$$
(t_0, T_{t_0}) \mapsto \Phi_t(T_{t_0}) = T_{t_0 + t}
$$

is a bijection for all $t$. Indeed, let $T_{t_0}$ be fixed and $t_0$, $t_1$ be two timesteps; thanks to the existence and unicity of the solution to the ODE with this initial condition: $\Phi_{t_0 + t_1} = \Phi_{t_0} \circ \Phi_{t_1} = \Phi_{t_1} \circ \Phi_{t_0}$. Therefore, $\Phi_t$ is a bijection and $\Phi_t^{-1} = \Phi_{-t}$. Moreover, the flow is differentiable if $f$ is continuously differentiable as well, which is not a restrictive assumption if it is implemented by a neural network with differentiable activation functions.

#### C.2.2   PRESERVATION OF MUTUAL INFORMATION BY INVERTIBLE MAPPINGS

A proof of the following result is given by Kraskov et al. (2004). We indicate below the major steps of the proof. Let $X$ and $Y$ be two random variables with marginal densities $\mu_X$, $\mu_Y$. Let $F$ be a

diffeomorphism acting on $Y$, $Y' = F(Y)$. If $J_F$ is the determinant of the Jacobian of $F$, we have:

$$\mu'(x, y') = \mu(x, y)J_F(y').$$

Then, expressing the mutual information $I$ in integral form, with the change of variables $y' = F(y)$ ($F$ being a diffeomorphism), results in:

$$I(X, Y') = \iint_{x,y'} \mu'(x, y') \log \frac{\mu'(x, y')}{\mu_X(x) \times \mu_{Y'}(y')} \, dx \, dy'$$
$$= \iint_{x,y} \mu(x, y) \log \frac{\mu(x, y)}{\mu_X(x) \times \mu_Y(y)} \, dx \, dy$$
$$I(X, Y') = I(X, Y).$$

## C.3 ENSURING DISENTANGLEMENT AT ANY TIME

As noted by Chen et al. (2016) and Achille & Soatto (2018), mutual information $I$ is a key metric to evaluate disentanglement. We show that our model logically preserves the mutual information between $S$ and $T$ through time thanks to the flow of the learned ODE on $T$. Indeed, with the result of mutual information preservation by diffeomorphisms, and $\Phi_t$ being a diffeomorphism as demonstrated above, we have, for all $t$ and $t'$:

$$I(S, T_t) = I(X, \Phi_{t'-t}(T_t)) = I(S, T_{t'}). \tag{29}$$

Hence, if $S$ and $T_t$ are disentangled, then so are $S$ and $T_{t'}$.

The flow $\Phi_t$ being dicretized in practice, its invertibility can no longer be guaranteed in general. Some numerical schemes (Chen et al., 2020) or residual networks with Lipschitz-constrained residual blocks (Behrmann et al., 2019) provide sufficient conditions to concretely reach this invertibility. In our case, we did not observe the need to enforce invertibility. We can also leverage the data processing inequality to show that, for any $t \geq t_0$:

$$I(S, T_{t_0}) \geq I(S, T_t), \tag{30}$$

since $T_t$ is a deterministic function of $T_{t_0}$. Since we constrain the very first $T$ value $T_{t_0}$ (i.e., we do not need to go back in time), there is no imperative need to enforce the invertibility of $\Phi_t$ in practice: the inequality also implies that, if $S$ and $T_{t_0}$ are disentangled, then so are $S$ and $T_t$ for $t \geq t_0$. Nevertheless, should the need to disentangle for $t < t_0$ appear, the aforementioned mutual information conservation properties could allow, with further practical work to ensure the effective invertibility of $\Phi_t$, to still regularize $T_{t_0}$ only. This is, however, out of the scope of this paper.

## D DATASETS

### D.1 WAVEEQ AND WAVEEQ-100

These datasets are based on the two-dimensional wave equation on a functional $w(x, y, t)$:

$$\frac{\partial^2 w}{\partial t^2} = c^2 \nabla^2 w + f(x, y, t), \tag{31}$$

where $\nabla^2$ is the Laplacian operator, $c$ denotes the wave celerity, and $f$ is an arbitrary time-dependent source term. It has several application in physics, modeling a wide range of phenomena ranging from mechanical oscillations to electromagnetism. Note that the homogeneous equation, where $f = 0$, admits product-separable solutions.

We build the WaveEq dataset by solving Equation (31) for $t \in [0, 0.298]$ and $x, y \in [0, 63]$. Sequences are generated using $c$ drawn uniformly at random in $[300, 400]$ for each sequence to imitate the propagation of acoustic waves, with initial and Neumann boundary conditions:

$$w(x, y, 0) = w(0, 0, t) = w(32, 32, t) = 0, \tag{32}$$

and, following Saha et al. (2020), we make use of the following source term:

$$f(x, y, t) = \begin{cases} f_0 e^{-\frac{t}{T_0}} & \text{if } (x, y) \in \mathcal{B}((32, 32), 5) \\ 0 & \text{otherwise} \end{cases}, \tag{33}$$

with $T_0 = 0.05$ and $f_0 \sim \mathcal{U}\big([1, 30]\big)$. The source term is taken non-null in a circular central zone only in order to avoid numerical differentiation problems in the case of a punctual source.

We generate 300 sequences of $64 \times 64$ frames of length 150 from this setting by assimilating pixel $(i, j) \in [\![0, 63]\!] \times [\![0, 63]\!]$ to a point $(x, y) \in [0, 63] \times [0, 63]$ and selecting a frame per time interval of size 0.002. This discretization is used to solve Equation (31) as its spatial derivatives are estimated thanks to finite differences; once computed, they are used in an ODE numerical solver to solve Equation (31) on $t$. Spatial derivatives are estimated with finite differences of order 5, and the ODE solver is the fourth-order Runge-Kutta method with the $3/8$ rule (Kutta, 1901; Hairer et al., 1993) and step size 0.001. The data are finally normalized following a min-max $[0, 1]$ scaling per sequence.

The dataset is then split into training (240 sequences) and testing (60 sequences) sets. Sequences sampled during training are random chuncks of length $\nu + 1 = 25$, including $\tau + 1 = 5$ conditioning frames, of full-size training sequences. Sequences used during testing are all possible chunks of length $\tau + 1 + 40 = 45$ from full-size testing sequences.

Finally, WaveEq-100 is created from WaveEq by selecting 100 pixels uniformly at random. The extracted pixels are selected before training and are fixed for both training and testing. Therefore, train and test sequences for WaveEq-100 consist of vector of size 100 extracted from WaveEq frames. Training and testing sequences are chosen to be the same as those of WaveEq.

### D.2    SEA SURFACE TEMPERATURE

SST is composed of sea surface temperatures of the Atlantic ocean generated using E.U. Copernicus Marine Service Information thanks to the state-of-the-art simulation engine NEMO. The use of a so-called reanalysis procedure implies that these data accurately represent the actual temperature measures. For more information, we refer to the complete description of the data by de Bézenac et al. (2018). The data history of this engine is available online.[2] Unfortunately, due to recent maintenance, data history is limited to the last three years; prior histories should be manually requested.

The dataset uses daily temperature acquisitions from Thursday $28^{\text{th}}$ December, 2006 to Wednesday $5^{\text{th}}$ April, 2017 of a $481 \times 781$ zone, from which 29 zones of size $64 \times 64$ zones are extracted. We follow the same setting as de Bézenac et al. (2018) by training all models with $\tau + 1 = 4$ conditioning steps and $\nu - \tau = 6$ steps to predict, and evaluating them only on zones 17 to 20. These zones are particularly interesting since they are the places where cold waters meet warm waters, inducing more pronounced motion.

We normalize the data in the same manner as de Bézenac et al. (2018). Each daily acquisition of a zone is first normalized using the mean and standard deviation of measured temperatures in this zone computed for all days with the same date of the year from the available data (daily history climatological normalization). Each zone is then normalized so that the mean and variance over all acquisitions correspond to those of a standard Gaussian distribution. These normalized data are finally fed to the model; MSE scores reported in Table 1 are computed once the performed normalization of the data and model prediction is reverted to the original temperature measurement space, in order to compute physically meaningful scores.

Training sequences correspond to randomly selected chunks of length $\nu = 10$ in the first 2987 acquisitions (corresponding to $80\%$ of total acquisitions), and testing sequences to all possible chunks of length $\nu = 10$ in the remaining 747 acquisitions.

### D.3    MOVING MNIST

This dataset involves two MNIST digits (LeCun et al., 1998) of size $28 \times 28$ that linearly move within $64 \times 64$ frames and deterministically bounce against frame borders following reflection laws. We use the modified version of the dataset proposed by Franceschi et al. (2020) instead of the original one (Srivastava et al., 2015). We train all models in the same setting as Denton & Birodkar (2017), with $\tau + 1 = 5$ conditioning frames and $\nu - \tau = 10$ frames to predict, and test them to predict either 10 or 95 frames ahead. Training data consist in trajectories of digits from the MNIST training set, randomly generated on the fly during training. Test data are produced by computing a trajectory for

---

[2]https://resources.marine.copernicus.eu/?option=com_csw&view=details&
product_id=GLOBAL_ANALYSIS_FORECAST_PHY_001_024.

each digit of the MNIST testing set, and randomly pairwise combining them, thus producing 5000 sequences.

To evaluate disentanglement with content swapping, we report PSNR and SSIM metrics between the swapped sequence produced by our model and a ground truth. However, having two digits in the image, there is an ambiguity as to in which order target digits should be swapped in the ground truth. To account for this ambiguity and thanks to the synthetic nature of the dataset, we instead build two ground truth sequences for both possible digit swap permutations, and report the lowest metric between the generated sequence and both ground truths (i.e., we choose the closest ground truth to compare to with respect to the considered metric).

### D.4 3D WAREHOUSE CHAIRS

This multi-view dataset introduced by Aubry et al. (2014) contains 1393 three-dimensional models of chairs seen under the same periodic angles. We resize the original $600 \times 600$ images by center-cropping them to $400 \times 400$ images, and downsample them to $64 \times 64$ frames using the Lanczos filter of the Pillow library.[3]

We create sequences from this dataset for our model by assembling the views of each chair to simulate its rotation from right to left until it reaches its initial position. This process is repeated for each existing angle to serve as initial position for all chairs. We chose this dataset instead of Denton & Birodkar (2017)'s multi-view chairs dataset because the latter contains too few objects to allow both tested methods to generalize on the testing set, preventing us to draw any conclusion from the experiment. We train models on this dataset with $\tau + 1 = 5$ conditioning frames and $\nu - \tau = 10$ frames to predict, and test them to predict 15 frames within the content swap experiment. Training and testing data are constituted by randomly selecting 85% of the chairs for training and 15% of the remaining ones for testing. Disentanglement metrics are computed similarly to the ones on Moving MNIST, but with only one reference ground truth corresponding to the chair given as content input at the position of the chair given as dynamic input.

### D.5 TAXIBJ

This crowd flow dataset provided by Zhang et al. (2017) consists in two-channel $32 \times 32$ frames representing the inflow and outflow of taxis in Beijing, each pixel corresponding to a square region of the city. Observations are registered every thirty minutes. It is highly structured as the flows are dependent on the infrastructure of the city, and complex since methods have to account for non-local dependencies and model subtle changes in the evolution of the flows.

We follow the preprocessing steps of Wang et al. (2018) and Le Guen & Thome (2020) by performing a min-max normalization of the data to the $[0, 1]$ range. We train models on this dataset with $\tau + 1 = 4$ conditioning frames and $\nu - \tau = 4$ frames to predict, and test them to predict 4 frames like our competitors on the last four weeks of data which are excluded from the training set. MSE on this dataset is reported in the $[0, 1]$-normalized space and multiplied by a hundred times the dimensionality of a frame, i.e. by $100 \times 32 \times 32 \times 2$.

## E TRAINING DETAILS

Along with the code in the supplementary material, we provide in this section sufficient details in order to replicate our results.

### E.1 REPRODUCTION OF BASELINES

**PKnl.** We retrained PKnl (de Bézenac et al., 2018) on SST using the official implementation and the indicated hyperparameters.

**SVG, MIM and DDPAE.** We trained SVG (Denton & Fergus, 2018), MIM (Wang et al., 2019b) and DDPAE (Hsieh et al., 2018) on our version of Moving MNIST using the official implementation and the same hyperparameters that the authors used for the original version of Moving MNIST.

---

[3] https://pillow.readthedocs.io/

We trained MIM on SST using the recommended hyperparameters of the authors, and SVG by retaining the same hyperparameters as those used on KTH.

**DrNet.** We trained DrNet (Denton & Birodkar, 2017) on our version of Moving MNIST using the same hyperparameters originally used for the alternative version of the dataset on which it was originally trained (with digits of different colors). To this end, we reimplemented the official Lua implementation into a Python code in order to train it with a more recent infrastucture. We also trained DrNet on 3D Warehouse Chairs using the same hyperparameters used by its authors on the smaller multi-view chairs dataset on which they trained their method.

**PhyDNet.** We trained PhyDNet (Le Guen & Thome, 2020) on SST and our version of Moving MNIST using the official implementation and the same hyperparameters that the authors used for SST and the original version of Moving MNIST. We removed the skip connections used by the authors on the Moving MNIST dataset in order to perform a fairer comparison with other models, such as ours, in our experimental study that do not incorporate skip connections on this dataset.

### E.2 MODEL SPECIFICATIONS

#### E.2.1 IMPLEMENTATION

We used Python 3.8.1 and PyTorch 1.4.0 (Paszke et al., 2019) to implement our model. Each model was trained on an Nvidia GPU with CUDA 10.1. Training is done with mixed-precision training (Micikevicius et al., 2018) thanks to the Apex library.[4]

#### E.2.2 ARCHITECTURE

**Combination of $S$ and $T$.** As explained in Section 4, the default choice of combination of $S$ and $T$ as decoder inputs is the concatenation of both vectorial variables: it is generic, and allows the decoder to learn an appropriate combination function $\zeta$ as in Equation (4).

Nonetheless, further knowledge of the studied dataset can help to narrow the choices of combination functions. Indeed, we choose to multiply $S$ and $T$ before giving them as input to the decoder for both datasets WaveEq and WaveEq-100, given the knowledge of the existence of product-separable solutions to the homogeneous version of equation (i.e., without source). This shows that it is possible to change the combination function of $S$ and $T$, and that existing combination functions in the PDE literature could be leveraged for other datasets.

**Encoders $E_S$ and $E_T$, and decoder $D$.** For WaveEq, the encoder and decoder outputs are considered to be vectors; images are thus flattened before encoding and reshaped after decoding to $64 \times 64$ frames. The encoder is a MultiLayer Perceptron (MLP) with two hidden layers of size 1200 and internal ReLU activation functions. The decoder is an MLP with three hidden layers of size 1200, internal ReLU activation functions, and a final sigmoid activation function for the decoder. The encoder and decoder used for WaveEq-100 are similar to those used for WaveEq, but with two hidden layers each, of respective sizes 2400 and 150.

We used for SST a VGG16 architecture (Simonyan & Zisserman, 2015), mirrored between the encoder and the decoder, complemented with skip connections integrated into $S$ (Ronneberger et al., 2015) from all internal layers of the encoder to corresponding decoder layers, also leveraged by de Bézenac et al. (2018) in their PKnl model. We adapted this VGG16 architecture without skip connections for the $32 \times 32$ frames of TaxiBJ by removing the shallowest upsampling and downsampling operations in the VGG encoder and decoder. For Moving MNIST, the encoder and its mirrored decoder are shaped with the DCGAN discriminator and generator architecture (Radford et al., 2016), with an additional sigmoid activation after the very last layer of the decoder; this encoder and decoder DCGAN architecture is also used by DrNet and DDPAE. We highlight that we leveraged in both SST and Moving MNIST architectural choices that are also used in compared baselines, enabling fair comparisons.

For the two-dimensional latent space experiments on SST (see Appendix F.3), we use a modified version of the VGG encoder / decoder network by removing the two deepest maximum pooling layers,

---

[4]https://github.com/nvidia/apex.

thus preserving the two-dimensional latent structures. The decoder mirrors the encoder complemented with skip connections.

Regarding 3D Warehouse Chairs, we also followed the same architectural choices as DrNet with a ResNet18-like architecture for the encoders and a DCGAN architecture, followed by a sigmoid activation after the last layer for the decoder.

Encoders $E_S$ and $E_T$ taking as input multiple observations, we combine them by either concatenating them for the vectorial observations of WaveEq-100, or grouping them on the color channel dimensions for the other datasets where observations are frames. Each encoder and decoder layer was initialized from a normal distribution with standard deviation $0.02$ (except for biases initialized to $0$, and batch normalizations weights drawn from a Gaussian distribution with unit mean and a standard deviation of $0.02$).

**ODE solver.** Following the recent line of work assimilating residual networks (He et al., 2016) with ODE solvers (Lu et al., 2018; Chen et al., 2018), we use a residual network as an integrator for Equation (7). This residual network is composed of a given number $K$ of residual blocks, each block $i \in [\![1, K]\!]$ implementing the application $\mathrm{id} + g_i$, where $g_i$ is an MLP with a two hidden layers of size $H$ and internal ReLU activation functions. The parameter values for each dataset are:

- WaveEq and WaveEq-100: $K = 3$ and $H = 512$;
- SST (with linear latent states): $K = 3$ and $H = 1024$;
- Moving MNIST, 3D Warehouse Chairs and TaxiBJ: $K = 1$ and $H = 512$.

Each MLP is orthogonally initialized with the following gain for each dataset:

- WaveEq, WaveEq-100, SST (with linear latent states), 3D Warehouse Chairs and TaxiBJ: $0.71$;
- Moving MNIST: $1.41$.

For SST with two-dimensional states, the MLPs are replaced by convolutional layers with kernel size 3, padding 1 and a number of hidden channels equal to $H = 128$. We set $K = 2$ and an orthogonal initialization gain of $0.2$. ReLU activations are replaced by Leaky ReLU activations and preceded by batch normalization layers.

**Latent variable sizes.** $S$ and $T$ have the following vectorial dimensions for each dataset:

- WaveEq and WaveEq-100: $32$;
- SST, respectively $196 \times 16 \times 16$ and $64 \times 16 \times 16$; for the linear version, both are set to $256$.
- Moving MNIST and TaxiBJ: respectively, $128$ and $20$;
- 3D Warehouse Chairs: respectively, $128$ and $10$.

Note that, in order to perform fair comparisons, the size of $T$ for baselines without static component $S$ is chosen to be the sum of the vectorial sizes of $S$ and $T$ in the full model. The skip connections of $S$ for SST cannot, however, be integrated into $T$, as its evolution is only modeled in the latent space, and it is out of the scope of this paper to leverage low-level dynamics.

### E.3 Optimization

Optimization is performed using the Adam optimizer (Kingma & Ba, 2015) with initial learning rate $4 \times 10^{-4}$ for WaveEq, WaveEq-100, Moving MNIST, 3D Warehouse Chairs and SST and $4 \times 10^{-5}$ for TaxiBJ, and with decay rates $\beta_1 = 0.9$ (except for the experiments on Moving MNIST where we choose $\beta_1 = 0.5$) and $\beta_2 = 0.99$.

**Loss function.** Chosen coefficients values of $\lambda_{\mathrm{pred}}$, $\lambda_{\mathrm{AE}}$, $\lambda_{\mathrm{reg}}^S$, and $\lambda_{\mathrm{reg}}^T$ are the following:

- $\lambda_{\mathrm{pred}} = 45$;

Table 4: Prediction and content swap PSNR and SSIM scores of variants of our model.

| Models | Pred. $(t+10)$ | | Pred. $(t+95)$ | | Swap $(t+10)$ | | Swap $(t+95)$ | |
|---|---|---|---|---|---|---|---|---|
| | PSNR | SSIM | PSNR | SSIM | PSNR | SSIM | PSNR | SSIM |
| Ours | **21.70** | **0.9088** | **17.50** | **0.7990** | **18.42** | **0.8368** | **16.50** | **0.7713** |
| Ours (without $S$) | 20.46 | 0.8867 | 14.95 | 0.6707 | — | — | — | — |
| Ours ($\lambda_{\text{AE}} = 0$) | 21.61 | 0.9058 | 16.58 | 0.7611 | 18.21 | 0.8309 | 15.79 | 0.7399 |
| Ours ($\lambda_{\text{reg}}^{S} = 0$) | 15.99 | 0.6900 | 12.31 | 0.5702 | 13.73 | 0.5476 | 12.07 | 0.5556 |
| Ours ($\lambda_{\text{reg}}^{T} = 0$) | 15.63 | 0.7369 | 14.02 | 0.7253 | 14.91 | 0.7154 | 13.95 | 0.7234 |
| Ours (GRU) | 21.66 | 0.9088 | 15.45 | 0.4888 | 17.70 | 0.8178 | 14.77 | 0.4718 |

- $\lambda_{\text{AE}} = 45$ for TaxiBJ; 10 for SST (linear)and Moving MNIST; 1 for WaveEq, WaveEq-100 and 3D Warehouse Chairs; 0.1 for SST;

- $\lambda_{\text{reg}}^{S} = 100$ for SST; $\lambda_{\text{reg}}^{S} = 45$ for WaveEq, WaveEq-100, SST (linear) and Moving MNIST; 1 for 3D Warehouse Chairs; 0.0001 for TaxiBJ;

- $\lambda_{\text{reg}}^{T} = \frac{1}{2}p \times 10^{-3}$ for WaveEq, WaveEq-100, Moving MNIST, 3D Warehouse Chairs and TaxiBJ (where $p$ is the dimension of $T$); $\frac{1}{2}p \times 10^{-2}$ for SST (linear); $5 \times 10^{-6}$ for SST.

The batch size is chosen to be 128 for WaveEq, WaveEq-100, Moving MNIST and 3D Warehouse Chairs, and 100 for SST and TaxiBJ.

**Training length.** The number of training epochs for each dataset is:

- WaveEq and WaveEq-100: 250 epochs;

- SST: 30 epochs; SST (linear): 80 epochs;

- Moving MNIST: 800 epochs, with an epoch corresponding to $200\,000$ trajectories (the dataset being infinite), and with the learning rate successively divided by 2 at epochs 300, 400, 500, 600, and 700;

- 3D Warehouse Chairs: 120 epochs;

- TaxiBJ: 550 epochs, with the learning rate divided by 5 at epochs 250, 300, 350, 400 and 450.

### E.4 PREDICTION OFFSET FOR SST

Using the formalism of our work, our algorithm trains to reconstruct $v = (v_{t_0}, \ldots, v_{t_1})$ from conditioning frames $V_\tau(t_0)$. Therefore, it first learns to reconstruct $V_\tau(t_0)$.

However, the evolution of SST data is chaotic and predicting above an horizon of 6 with coherent and sharp estimations is challenging. Therefore, for the SST dataset only, we chose to supervise the prediction from $t = t_0 + (\tau + 1)\Delta t$, i.e, our algorithm trains to forecast $v_{t_0+(\tau+1)\Delta t}, \ldots, v_{t_1}$ from $V_\tau(t_0)$. It simply consists in making the temporal representation $E_T\left(V_\tau(t_0)\right)$ match the observation $v_{t_0+(\tau+1)\Delta t}$ instead of $v_{t_0}$. This index offset does not change our interpretation of spatiotemporal disentanglement through separation of variables.

## F ADDITIONAL RESULTS AND SAMPLES

### F.1 ABLATION STUDY ON MOVING MNIST

We report in Table 4 the results of an ablation study of our model on Moving MNIST, that we comment in Section 5.2.

Table 5: FVD score of compared models on KTH. The bold score indicates the best performing method.

| Ours | PhyDNet | SVG | DrNet |
|------|---------|-----|-------|
| **330** | 384 | 375 | 383 |

## F.2 PRELIMINARY RESULTS ON KTH

The application of our method to natural videos is an interesting perspective, but would motivate further adaptation of the model (see perspectives in the conclusion), in particular regarding the integration of stochastic dynamics. Indeed, there is a consensus in the literature (e.g.: Denton & Fergus (2018); Villegas et al. (2019); Weissenborn et al. (2020)) indicating that human motion datasets require stochastic modeling because of the inherently highly random events occurring in these videos. Tackling this issue would require to incorporate stochasticity in our model, for example leveraging variational autoencoders like Denton & Fergus (2018), or supplement it with adversarial losses on the image space, for instance like Mathieu et al. (2016) and Lee et al. (2018). These changes are feasible, but are out of the scope of this paper.

Nonetheless, we investigate the realistic video dataset KTH (Schüldt et al., 2004), which is an action recognition video database featuring various subjects performing actions in front of different backgrounds. We trained our model, SVG, DrNet and PhyDNet on this dataset. DrNet and PhyDNet are powerful deterministic approaches, while SVG is a standard stochastic video prediction model. We compare all models in terms of FVD (Unterthiner et al., 2018, lower is better), which is a metric based on deep features that evaluates the realism of the generated videos.

Results are reported in Table 5. We observe that our model substantially outperforms the considered baselines. These significant results against powerful deterministic baselines, and even the standard stochastic method SVG, confirm our advantage at modeling complex dynamics and support our claim that our model lays the foundations for domain-specific methods, such as a stochastic version for natural videos.

**Reproductibility.** We use the following training parameters for KTH:

- we follow the same dataset processing and evaluation procedure as Denton & Fergus (2018);
- we train our model on 125 epochs with batch size 100, with an epoch being defined as 100 000 training sequences;
- we set the learning rate to $2 \times 10^{-4}$ and the same optimizer parameters as for SST;
- $\lambda_{\mathrm{pred}} = 45$, $\lambda_{\mathrm{AE}} = 10 = \lambda_{\mathrm{reg}}^{S} = 10$, $\lambda_{\mathrm{reg}}^{T} = p \times 10^{-4}$;
- the size of $S$ and $T$ are respectively 128 and 50;
- the ODE is solved with a flat latent architecture and parameters $K = 1$ and $H = 512$;
- the encoder and decoder architecture is VGG16 with skip connections integrated into $S$ from $E_S$ to $D$, and with the decoder output being given to a final sigmoid activation.

We reproduced SVG, DrNet and PhyDNet using the recommended hyperparameters of their authors. We trained PhyDNet for 125 epochs, like our model, to obtain a fair evaluation despite its low efficiency (six times slower than ours).

## F.3 MODELING SST WITH SEPARATION OF VARIABLES

We present in Table 6 results of Table 1 for SST, complemented with an alternative version of our model obtained using vectorial representation for $S$ and $T$ and MLPs to compute the derivative of $T$. The latter setting corresponds to a strictly enforced separation of spatial and dynamical variables, with results significantly outperforming powerful methods PhyDNet, PKnl and SVG thanks to this separation, as attested by the corresponding ablation without a static component.

Table 6:   Forecasting performance on SST of PKnl, PhyDNet and our model with respect to indicated prediction horizons. Bold scores indicate the best performing method.

| Models | MSE | | SSIM | |
|---|---|---|---|---|
| | $t+6$ | $t+10$ | $t+6$ | $t+10$ |
| PKnl | 1.28 | 2.03 | 0.6686 | 0.5844 |
| PhyDNet | 1.27 | 1.91 | 0.5782 | 0.4645 |
| SVG | 1.51 | 2.06 | 0.6259 | 0.5595 |
| MIM | 0.91 | 1.45 | 0.7406 | 0.6525 |
| Ours | **0.86** | **1.43** | **0.7466** | **0.6577** |
| Ours (without $S$) | 0.95 | 1.50 | 0.7204 | 0.6446 |
| Ours (linear) | 1.15 | 1.80 | 0.6837 | 0.5984 |
| Ours (linear, without $S$) | 1.46 | 2.19 | 0.6200 | 0.5456 |

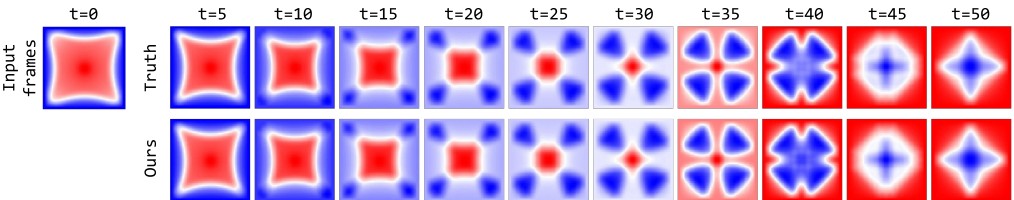

Figure 6: Example of predictions of our model on WaveEq.

However, sea surface temperature exhibits highly local structure that can be assimilated to a flow in a coarse approximation. For example, there is transport of large bodies of hot and cold water. Accordingly, performances may be enhanced by considering local dependencies in the dynamics, as also implemented by MIM and PhyDNet. We propose to do so by considering like the latter methods two-dimensional latent states for the static $S$ and the dynamical $T$, and convolutional networks to model the derivative of $T$.

Accounting for such locality in the dynamics amounts to implementing another separation than the usual separation between $t$ and spatial variables. Indeed, it rather excludes unknown content variables from the dynamics. The resulting dynamics is then a PDE over time $t$ and the observation coordinates $x$ and $y$ that we implement using convolutional neural networks, following Long et al. (2018) and Ayed et al. (2020). This different kind of separation of variables simplifies learning by estimating a PDE that is simpler than the original one, since it acts on fewer variables. It highlights the generality of our intuition of using the separation of variables, which may be used in other settings that strict spatiotemporal disentanglement. This approach, while still maintaining disentangling properties, significantly improves prediction performances.

Note that our proposition remains computationally much lighter than the alternatives MIM, PhyDNet and SVG.

## F.4   ADDITIONAL SAMPLES

### F.4.1   WAVEEQ

We provide in Figure 6 a sample for the WaveEq dataset, highlighting the long-term consistency in the forecasts of our algorithm.

We also show in Figure 7 the effect in forecasting of changing the spatial code $S$ from the one of another sequence.

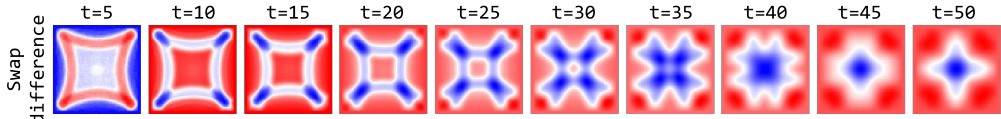

Figure 7: Evolution of the scaled difference between the forecast of a sequence and the same forecast with a spatial code coming from another sequence for the WaveEq dataset.

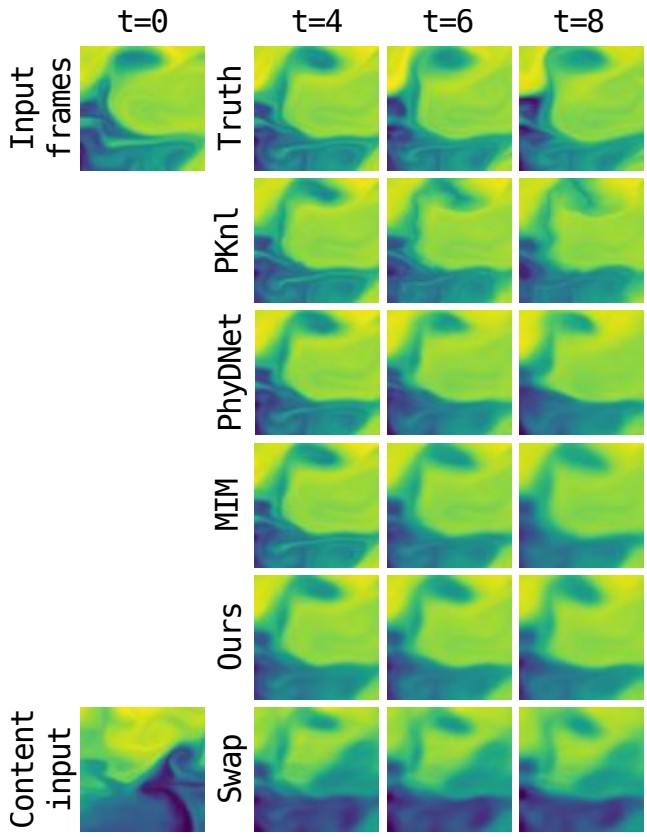

Figure 8: Example of predictions of compared models on SST.

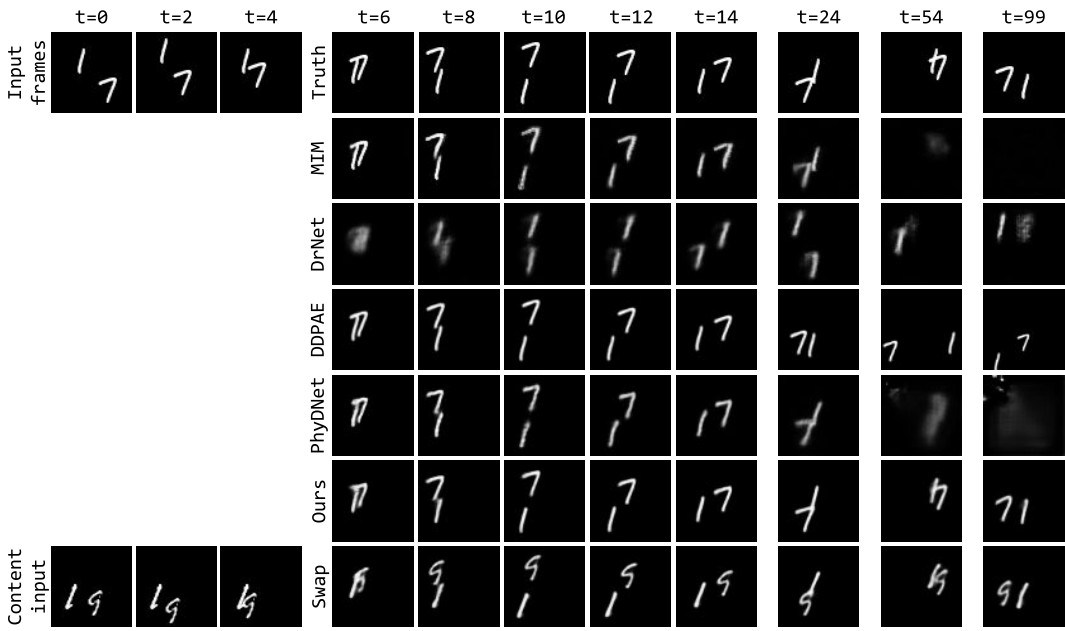

Figure 9: Example of predictions of compared models on Moving MNIST.

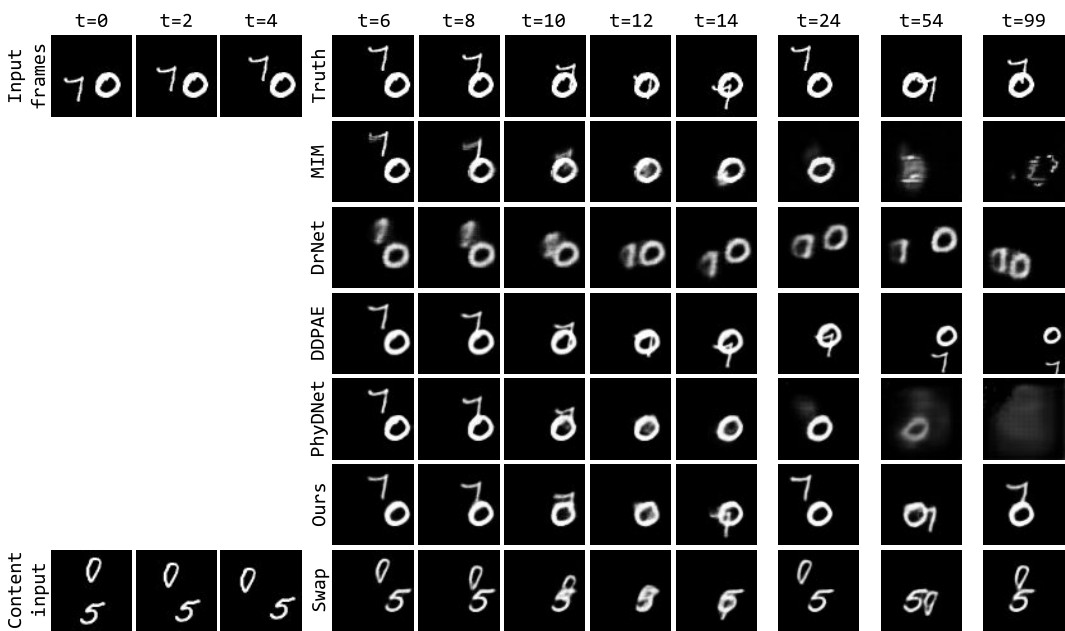

Figure 10: Example of predictions of compared models on Moving MNIST.

### F.4.2 SST

We provide an additional sample for SST in Figure 8.

### F.4.3 MOVING MNIST

We provide two additional samples for Moving MNIST in Figures 9 and 10.

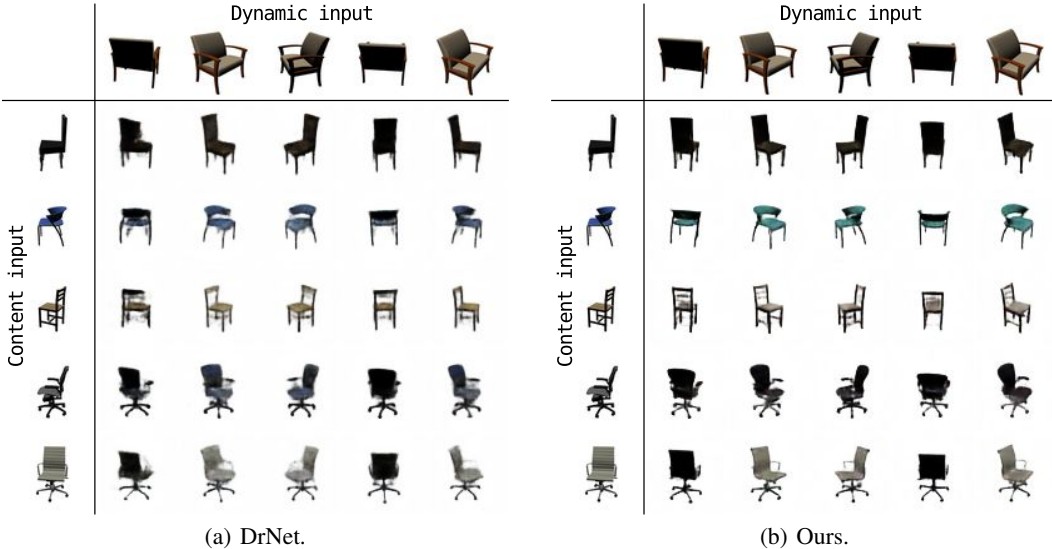

Figure 11: Fusion of content (first column) and dynamic (first row) variables in DrNet and our model on 3D Warehouse Chairs.

### F.4.4 3D WAREHOUSE CHAIRS

We provide a qualitative comparison for the content swap experiment between our model and DrNet for 3D Warehouse Chairs in Figure 11. We notice that DrNet produces substantially more blurry samples than our model and has difficulties to capture the exact dynamic of the chairs.

