# OpenReview forum: "PDE-Driven Spatiotemporal Disentanglement"
_ICLR.cc/2021/Conference — ICLR 2021 Poster_

### Official Review · AnonReviewer4 · 2020-10-28

**Rating:** 7
**Confidence:** 3

**Review:**

The paper presents a spatiotemporal disentanglement method for handling sequence data. Solving high-dimensional PDEs, for deriving the exact dynamics, is difficult; hence, this work proposes learning time-invariant and time-dependent representations separately to solve this problem. To achieve this goal, the authors devised a model that incorporates a temporal ODE process. The provided experiments indicate that the method achieves good performance, although the gain is not consistent.

The overall derivation and methodology of this paper are technically sound, and I guess discarding $S^\prime$ was a practical choice for learning time-invariant representation. Although the proposed model is generally applicable, the experiments only cover prediction results on classical image datasets. Nevertheless, the underlying timescales of datasets are sufficiently varied to demonstrate the effectiveness in general temporal sequences.

[Quality]

The paper is clearly written overall. However, Section 4 was a bit hard to follow, which is the core part of this paper. For example, "the system coordinates" and reasons for the relaxation (Sec. 4.2) were not clearly explained for people who are not familiar with the area. It would be harder to understand the architecture if Fig. 1 was not given.

[Originality]

The originality of the paper is not stellar, but sufficient for acceptance.

[Significance]

The significance of this work is mainly for model architecture. I believe these kinds of approaches, which can internally model continuous dynamics, are heavily preferred when solving real-world problems. Therefore, the significance of the paper is sufficient for acceptance.

---

> ### Author Response · Authors · 2020-11-13
> **Answer to Reviewer #4**
>
> We would like to thank you for your positive review.
>
>
> ## Clarifications
>
> We develop in the following the unclear points that you mentioned. We will make sure to clarify them in the updated manuscript following your feedback. Please let us know if you have any additional questions; we would gladly answer them.
>
> When formulating a PDE, the system coordinates for space (x, y, etc.) and time (t) naturally appear as variables of the solution. Accessing spatial coordinates may be either costly or infeasible when the partial observation of the phenomenon obfuscates the knowledge of such coordinates. For example, there can be numerous unknown variables in SST that drive the evolution of the observed phenomenon, such as the temperature of the water at large depths; without prior knowledge on this phenomenon, the nature and number of these variables are unknown. Non-physical datasets such as Moving MNIST and 3D Warehouse Chairs are partially observed as well and then also share this issue: for instance, the observed chairs are only 2D projections of 3D objects. Therefore, we chose not to model the solution of a PDE as traditionally performed in parametric PDE models, i.e. as a function of the time t and space variables x, y. The dynamics in our model should then be an explicit function of t only.
>
> Henceforth, the described relaxation consists in eliminating the explicit modeling of spatial coordinates by learning dynamic and time-invariant representations accounting respectively for the time-dependent and space-dependent parts of the solution. The time-invariant representation is then supposed to learn the unknown spatial components of the system in a single vector, thus removing these unknowns from the model. This is summarized by Eq. (5) and Eq. (6).
>
>
> ## Experimental Results
>
> Our model achieves state-of-the-art results on all considered tasks and datasets, both regarding disentanglement and prediction.
>
> The only evaluation on which our model falls behind some baselines is short-term prediction for Moving MNIST. Nevertheless, short-term Moving MNIST predictions do not show that the dynamic is well-learned but rather accounts for encoding and decoding capacities of the model. In contrast, as analyzed in Section 5.2, long-term prediction results evidence that our model learns more consistent and disentangled dynamics, unlike all baselines.

---

> ### Author Response · Authors · 2020-11-25
> **Revision**
>
> We provide in our updated submission more in-depth explanations for the unclear parts that you mentioned in Section 4.2.
>
> We strengthen our experimental results to highlight the relevance of our approach by including an RNN ablation and a new real-world dataset (TaxiBJ [1]). We finally improve our results on SST and KTH.
>
> [1] Deep Spatio-Temporal Residual Networks for Citywide Crowd Flows Prediction. AAAI 2017.

---

### Official Review · AnonReviewer3 · 2020-10-28
**Another generative model for videos with a slow feature loss**

**Rating:** 5
**Confidence:** 4

**Review:**

The authors present a generative model for videos where the latent trajectories have two components - a term without a slowness loss that represents "content" and a term with a slowness loss that represents "style". They present results on a dataset simulating the wave equation and on videos of moving MNIST digits and 3D chairs. The results are generally good, especially for long roll-outs, and they demonstrate something like disentangling by showing that the identities of the digits can be swapped in the moving MNIST data.

My main objection with the paper is that it has nothing to do with PDEs or separation of variables. The actual latent trajectories are simulated as *ODEs*, not PDEs, which are then used to generate images. The justification in terms of separation of variables is also a bait-and-switch...a slowness penalty is added to the loss for one of the latent trajectories, that is all. Factored latent trajectories are a well-established modeling technique for time series already. The use of ODEs parameterized by neural networks rather than, say, LSTMs or other RNN architectures is also more a difference in degree than in kind from other sequence models. Much like how ResNets become neural ODEs in the limit of very deep networks, simply using Runge-Kutta updates parameterized by a neural network is still technically a kind of RNN (if you define RNN very loosely as any nonlinear iterated function learnable by gradient descent). So I'm not sure that this paper actually does most of what it claims to be doing in the motivation.

---

> ### Author Response · Authors · 2020-11-13
> **Answer to Reviewer #3 (1/2)**
>
> We thank you for your review, but respectfully disagree on your main conclusions. We are sorry to read that you believe that our model is falsely advertised and non significant. We expose in the following factual counterarguments and hope to engage in a constructive discussion on this matter.
>
>
> # PDEs and Separation of Variables
>
>
> We assume for modeling purposes that our observations are driven by PDEs, as discussed in Sections 1, 4.1 and C.1, but the model itself does not implement PDEs: this is a key point of the paper, driven by the separation of variables. We would like to point out that we never claimed to implement PDEs in our model. We detail this issue below.
>
> We stated in all our work, including the title and abstract, that PDEs and the separation of variables were considered along the paper as principled tools that drive our approach to derive appropriate cost functions. To the best of our knowledge, this work is the first one to interpret spatiotemporal disentanglement as a separation of variables in a PDE. Because it provides state-of-the-art results on several datasets, we believe that our interpretation may also be used in other settings, for example with fully observed phenomena which would not necessarily require any relaxation.
>
> As duly noted by the reviewer and in the manuscript, we simulate latent trajectories as ODEs. This is exactly the desired goal and flows directly from the application of the method of separation of variables, as explained in the following.
>
> Our modeling assumption is that the observed phenomenon is driven by an unknown PDE. However, learning numerical or analytical solutions to this PDE is notoriously complex. Therefore, we are looking for separable solutions to this PDE, that we interpret as spatiotemporal disentanglement. We then study how to apply such separation in the context of a partially observable phenomenon and thus formulate its consequences as cost functionals and inductive priors.
>
> Our derived formulation presents few major properties:
>   - It separates the learning of the solution into time-invariant and time-varying functions. It indeed concurs with prior approaches, as noted in our related work, and confirms our interpretation of spatiotemporal disentanglement as separation of variables.
>   - The separation of variables makes the time-varying component of the overall solution obeying an ODE, simplifying the integration and learning of the PDE over time enabled by recent advances [1, 2]. Without loss of generality, we model this time dependency as a vectorial first-order ODE. Besides their continuous nature, ODEs provide interesting modeling properties. For example, the flow of an ODE being a bijection, it preserves mutual information (Sections 4.4 and C.2, and Eq. (12)) which may explain our good long-term disentanglement performances (Table 2).
>   - Compared to previous disentanglement approaches involving complex learning tools such as latent adversarial discrimination for DrNet or factorized variational autoencoding for DDPAE, the effective constraints derived from our principled modeling choices only involve regression penalties. Their relevance is discussed in an ablation study (Section F.1, Table 3, analyzed in Section 5.2).
>
> We will deepen the discussion precising our motivations for leveraging the separation of variables in Sections 4.1 and 4.2.
>
>
> # Links and Comparisons with RNNs
>
> Much more than a difference in degree, leveraging ODEs instead of conventional LSTMs is a fundamental choice.
>
> Of course, ODE resolution schemes are recurrent. However, the expression “recurrent neural networks” refers in the machine learning community to architectures such as GRU, LSTM and ConvLSTM which are fundamentally different from classical state-space dynamical systems. We used this expression in this usual sense in our submission. Therefore, our claim based on prior work and our experimental results holds: ODE-based dynamics are more adapted to model spatiotemporal phenomena than conventional networks such as LSTMs.
>
> We reinforce our claim with additional experiments to be included in the updated submission. We reproduced our Moving MNIST experiments by replacing our latent ODE for the integration of T by a GRU, leading to the following results.
>
> |  | Prediction | (t+10) | Prediction | (t+95) | Disentangl. | (t+10) | Disentangl. | (t+95) |
> |-|:-:|:-:|:-:|:-:|:-:|:-:|:-:|:-:|
> | Models | PSNR | SSIM | PSNR | SSIM | PSNR | SSIM | PSNR | SSIM |
> | Ours (GRU) | 21.66 | 0.9088 | 15.45 | 0.4888 | 17.70 | 0.8178 | 14.77 | 0.4718 |
> | Ours | **21.76** | **0.9092** | **17.89** | **0.8094** | **18.37** | **0.8344** | **16.71** | **0.7777** |
>
> As expected, using a recurrent neural network instead of an ODE dramatically hurts the performance of the model, both in terms of prediction and disentanglement.
>
> The study of the good performances obtained with ODE-based trajectories is beyond the scope of this paper, but is an active research direction in the literature [2, 3, 4].

---

> > ### Author Response · Authors · 2020-11-13
> > **Answer to Reviewer #3 (2/2)**
> >
> > # Relationship to Prior Factored Latent Trajectories Methods
> >
> > We conducted an in-depth related work discussion in Section 2, accounting notably for prior spatiotemporally disentangled method. Furthermore, our method outperforms numerous and relevant baselines on several datasets. We would gladly include and discuss any forgotten relevant prior work that you might indicate in our related work section.
> >
> >
> > # References
> >
> > [1] Y. Lu et al. Beyond Finite Layer Neural Networks: Bridging Deep Architectures and Numerical Differential Equations. ICML 2018.
> > [2] R. T. Q. Chen et al. Neural Ordinary Differential Equations. NeurIPS 2018.
> > [3] C. Finlay et al. How to Train Your Neural ODE: the World of Jacobian and Kinetic Regularization. ICML 2020.
> > [4] H. A. Ayyubi et al. Progressive Growing Of Neural Odes. ICLR 2020 Workshop Integration of Deep Neural Models and Differential Equations.

---

> ### Author Response · Authors · 2020-11-25
> **Revision**
>
> As promised, we deepen in Sections 4.1, 4.2 and 4.3 of the updated submission the link between the proposed model and the separation of variables for PDEs. We also provide several significant experiments validating our claims. Notably, we include the ablation study on MNIST modeling the dynamics with GRU-RNN that validates our approach.

---

### Official Review · AnonReviewer1 · 2020-10-29
**Well-written paper with interesting but insufficient experiments**

**Rating:** 7
**Confidence:** 5

**Review:**

This paper proposes a new framework for spatiotemporal disentanglement. In particular, it contributes to the disentanglement of content and dynamics using neural ODEs.

Strength:
1. I think it is a well-written paper with interesting experimental design, especially the “swap” and “multi-view” experiments that measure the degree of disentanglement.
2. The proposed model consistently outperforms well-established approaches for video prediction, including DrNet, DDPAE, and PhyDNet, which is also a PDE-based model for spatiotemporal disentanglement.

Weakness:
1. Although the effectiveness of the model has been validated on several datasets, most of them are synthetic. I strongly encourage the authors to include real human motion datasets. For example, DrNet was evaluated on KTH, and PhyDNet was evaluated on Human3.6M.
2. The model was only compared with PKnl and PhyDNet on the Wave and SST datasets, and I am curious about how models beyond disentanglement, such as SVG and MIM, perform in this case. Also, the results in Figure 2 appear to be quite blurry and do not show significant improvement. These experiments mainly show the general ability of the proposed model for video prediction rather than spatiotemporal disentanglement.

Correctness:
In Eq. (12), the authors use a Gaussian prior to convey dynamic information and exclude the spatial information. Is it a good thing or a bad thing from the view of modeling temporal dynamics? A question is: can such a simple dynamic model be applied to datasets with complex motion information?

---

> ### Author Response · Authors · 2020-11-13
> **Answer to Reviewer #1 (1/2)**
>
> We would like to thank you for your constructive review. We address your concerns in a first answer below. We will keep you informed about additional experimental results and make sure to update our submission accordingly as soon as possible.
>
> # Choice of Datasets
>
> ## Synthetic Datasets
>
> Our choice of synthetic datasets Moving MNIST and 3D Warehouse Chairs is motivated by the necessity to be able to compute disentanglement metrics in controlled settings in order to assess the disentanglement abilities of compared models. These quantitative experiments were not considered in earlier works. Moreover, despite being synthetic, Moving MNIST (evaluating long-term accuracy in the prediction of a mechanical problem) and 3D Warehouse Chairs (involving complex shapes) are challenging for all models, hence their relevance in our comparison with the state of the art.
>
> ## Human Motions Datasets
>
> As mentioned in Appendix Section F.2 and in our conclusion, a consensus in the literature (e.g., [1, 2, 3]) indicates that human motion datasets require stochastic modeling because of the inherently highly random events occurring in these videos. Tackling this issue would require to incorporate stochasticity in our model, for example leveraging variational autoencoders like [1], or supplement it with adversarial losses on the image space, for instance like [4, 5]. These changes are feasible but are out of the scope of this paper. The goal of the present work is to find a principled spatiotemporal disentanglement model compared to prior complex deterministic approaches.
>
> Nonetheless, we already investigated the realistic video dataset KTH in Appendix Section F.2 with encouraging prediction performances matching the investigated baselines (SVG and DrNet). We will include PhyDNet in this comparison. These results obtained in our deterministic setting lay promising foundations for future models deploying more involved and domain-specific tools.
>
>
> # Additional Baselines and Results on Physical Datasets
>
> ## Disentanglement Evaluation of Physical Datasets
>
> As mentioned in the previous section of this answer, evaluating the disentanglement ability of models requires a controlled setting with prior knowledge on the spatiotemporal disentanglement in the dataset. Therefore, the only disentanglement studies that could be done on SST and WaveEq consisted in showing that such disentanglement indeed improves performances and qualitatively assessing that altering the invariant component of the model indeed changes predictions. We accordingly conducted these experiments on SST and WaveEq (on which we did not consider any comparison with prior work as it was only designed for introductory experiments).
>
> ## Additional Baselines
>
> We will add in the updated submission results on SST for MIM and SVG, which we did not include as they were not originally tested on this dataset. In the meantime, we highlight that PhyDNet is reported to outperform MIM and other baselines on this dataset [6]. Since we outperform PhyDNet, we assume the same upcoming results for MIM.
>
> ## Results Analysis on SST
>
> We stress that SST is a challenging dataset involving chaotic behavior even at short time scales and remains unsolved in the state of the art. These difficulties indeed lead to a slight blur in our predictions. However, our model shows the most accurate trajectories compared to PKnl and PhyDNet, as attested by their numerical performances with respect to MSE and SSIM. In particular, we outperform both baselines with respect to SSIM, which is a metric that penalizes blurred predictions. Qualitatively, we notice that PhyDNet produces much blurrier prediction than ours (cf. also Figure 4.c. in [6]) -- hence its low SSIM -- and PKnl, while sharper, consistently predicts wrong displacements.

---

> > ### Author Response · Authors · 2020-11-13
> > **Answer to Reviewer #1 (2/2)**
> >
> > # Correctness
> >
> > Your questions relate to several subjects. We try to answer them in the following.
> >
> > Imposing a Gaussian prior is a reasonable assumption in the model, like in dynamical variational models [1, 7] or for arbitrary distribution modeling [8]. We show in our ablation study of Appendix Section F.1 (Table 3), discussed in Section 5.2, that this Gaussian prior is key to achieve disentanglement as well as state-of-the-art prediction performances.
> >
> > More generally, a disentangled prediction system, besides offering interpretability, simplifies the dynamics by removing information that it does not need to account for. Our ablation study shows that considering spatial information in the dynamics by removing S (thus removing the Gaussian prior as well) significantly hurts the prediction abilities, validating the search for a constrained dynamic. This is also supported by prior approaches discussed in the related work which verified this hypothesis on various spatiotemporal phenomena.
> >
> > Moreover, the proposed dynamic is suited to model complex motions, as indicated by our results on SST. Vectorial representations for T with an ODE resolution scheme parameterized by neural networks ensure that our temporal model can approximate any continuous function. As an illustration, ODE-based models have already been successfully leveraged in the literature for prediction, density modeling and classification, e.g. in [6, 9, 10].
> >
> >
> > # References
> >
> > [1] E. Denton & R. Fergus. Stochastic Video Generation with a Learned Prior. ICML 2018.
> > [2] R. Villegas et al. High Fidelity Video Prediction with Large Stochastic Recurrent Neural Networks. NeurIPS 2019.
> > [3] D. Weissenborn et al. Scaling Autoregressive Video Models. ICLR 2020.
> > [4] M. Mathieu et al. Deep Multi-Scale Video Prediction Beyond Mean Square Error. ICLR 2016.
> > [5] A. X. Lee et al. Stochastic Adversarial Video Prediction. ArXiv. 2018.
> > [6] V. Le Guen & N. Thome. Disentangling Physical Dynamics from Unknown Factors for Unsupervised Video Prediction. CVPR 2020.
> > [7] J.-T. Hsieh et al. Learning to Decompose and Disentangle Representations for Video Prediction. NeurIPS 2018.
> > [8] W. Grathwohl et al. FFJORD: Free-Form Continuous Dynamics for Scalable Reversible Generative Models. ICLR 2019.
> > [9] I. Ayed et al. Learning the Spatio-Temporal Dynamics of Physical Processes from Partial Observations. ICASSP 2020.
> > [10] Y. Rubanova et al. Latent ODEs for Irregularly-Sampled Time Series. NeurIPS 2019.

---

> ### Author Response · Authors · 2020-11-25
> **Revision**
>
> As requested, we add in the updated manuscript the strong baselines MIM and SVG on SST, that we outperform. We improve our results on this dataset, hence producing sharper frames. We additionally extend the results analysis on this dataset.
>
> Along the updated submission, we show experimental results on three real-world datasets and three synthetic ones. The real-world datasets are diverse and cover different domains (physical phenomenon, video and traffic flows). Besides SST and KTH, we add the commonly used TaxiBJ [1] dataset, on which we achieve the state of the art. Moreover, we add improved experimental results and details on KTH in appendix F.2, evidencing better prediction performances than SVG, DrNet and PhyDNet.
>
> [1] Deep Spatio-Temporal Residual Networks for Citywide Crowd Flows Prediction. AAAI 2017.

---

### Author Response · Authors · 2020-11-13
**First Answers to Reviews**

We would like to thank the reviewers for their feedback. We individually responded to each review. We are looking forward to discussing with the reviewers about the raised questions and will make sure to update our submission to address your concerns and incorporate additional experimental results, including the ones we already provide in our answer to Reviewer #3.

---

### Author Response · Authors · 2020-11-24
**New and Improved Experimental Results**

We are pleased to share new experimental results that improve and expand the resultats that we presented in our initial submission. The submission was updated accordingly.


# KTH

We improve our performance on KTH by decreasing the FVD score from 372 to 330 (against 375 for SVG and 383 for DrNet). We also evaluate PhyDNet on this dataset, obtaining an FVD score of 384.

| Model |   Ours  | PhyDNet | SVG | DrNet |
|-------|:-------:|:-------:|:---:|:-----:|
| FVD   | **330** |   384   | 375 |  383  |

These significant results against powerful deterministic baselines, and even the standard stochastic method SVG, confirm our advantage at modeling complex dynamics and support our claim that our model lays the foundations for domain-specific methods, such as a stochastic version for natural videos.


# New Dataset: TaxiBJ

We consider the TaxiBJ dataset [1], which is a real-world spatiotemporal dataset representing trajectory flows of taxis in a 32x32 representation of Beijing, in our empirical evaluation.  We obtain state-of-the-art results on this dataset against several powerful baselines, as shown in the following table (with baseline results for PhyDNet and ConvLSTM reported from [2], and results for the other models taken from [3]).

| Model |    Ours   | Ours (w/o S) | PhyDNet |  MIM | E3D-LSTM [4] | Causal LSTM [5] | PredRNN [6] | ConvLSTM [7] |
|-------|:----:|:------------:|:-------:|:----:|:------------:|:---------------:|:-----------:|:------------:|
| MSE   | **39.8** |     41.7     |   41.9  | 42.9 |     43.2     |       44.8      |     46.4    |     48.5     |

Note that, similarly to other datasets, the version of our model without S is substantially less performant than the full model, showing the relevance of the proposed approach.


# SST

We improve the experimental study on SST in the following ways:
 - we now consider MIM and SVG as additional baselines that we trained on the dataset;
 - we drastically improve our experimental resultats, which allows us to maintain state-of-the-art performance.

The results are summarized in the following table.

|              |    MSE   |          |    SSIM    |            |
|--------------|:--------:|:--------:|:----------:|:----------:|
| Models       | t+6      | t+10     | t+6        | t+10       |
| PKnl         |   1.28   |   2.03   |   0.6686   |   0.5844   |
| PhyDNet      | 1.27     | 1.91     | 0.5782     | 0.4645     |
| SVG          | 1.51     | 2.06     | 0.6259     | 0.5595     |
| MIM          | 0.91     | 1.45     | 0.7406     | 0.6525     |
| Ours         | **0.86** | **1.43** | **0.7466** | **0.6577** |
| Ours (w/o S) | 0.95     | 1.50     | 0.7204     | 0.6446     |

We refer to the analysis of results in Section 5.1 for more details.


# References

[1] Deep Spatio-Temporal Residual Networks for Citywide Crowd Flows Prediction. AAAI 2017.
[2] V. Le Guen & N. Thome. Disentangling Physical Dynamics from Unknown Factors for Unsupervised Video Prediction. CVPR 2020.
[3] Y. Wang et al. Memory In Memory: A Predictive Neural Network for Learning Higher-Order Non-Stationarity from Spatiotemporal Dynamics. CVPR 2019.
[4] Y. Wang et al. Eidetic 3D LSTM: A Model for Video Prediction and Beyond. ICLR 2019.
[5] Y. Wang et al. PredRNN++: Towards A Resolution of the Deep-in-Time Dilemma in Spatiotemporal Predictive Learning. ICML 2018.
[6] Y. Wang et al. PredRNN: Recurrent Neural Networks for Predictive Learning Using Spatiotemporal LSTMs. NIPS 2017.
[7] X. Shi et al. Convolutional LSTM Network: A Machine Learning Approach for Precipitation Nowcasting. NIPS 2015.

---

### Author Response · Authors · 2020-11-25
**Revision**

We submitted a new version of our manuscript that addresses the reviewers’ comments and includes the new experimental results. We thank the reviewers for their feedback that allowed us to improve our submission and hope that it alleviates their concerns.

The main changes are the following:
 - we include several significant experimental results, as described in [this message](https://openreview.net/forum?id=vLaHRtHvfFp&noteId=veMXpEV1M9c);
 - we deepen and clarify the link between our model and the separation of variables method in Sections 4.1, 4.2 and 4.3;
 - we clarify the proposed relaxation in Section 4.2 following Reviewer #4’s comments;
 - we include an additional ablation study in Sections 5.2 and F.1 revealing the essential role of ODEs in the performance of our model by evidencing a performance drop when replacing ODEs with a standard GRU RNN;
 - we give additional results analysis and experimental details for SST and KTH.

---

### Decision · Program_Chairs · 2021-01-07
**Final Decision**

**Decision:**

Accept (Poster)

**Comment:**

This paper proposes a model for disentangling content and dynamics, but unlike the majority of previous work, the dynamics are modeled using ODEs rather than their discrete approximations - RNNs. The reviewers agree that the paper is well written, and the results look good, especially for longer trajectories. Hence, I am happy to recommend this paper for acceptance.